# Rule-based modulation of a sensorimotor transformation across cortical areas

Yi-Ting Chang[1,2], Eric A Finkel[1], Duo Xu[1,2], Daniel H O'Connor[1,2]*

[1]Solomon H. Snyder Department of Neuroscience, Kavli Neuroscience Discovery Institute, Brain Science Institute, Johns Hopkins University School of Medicine, Baltimore, United States; [2]Zanvyl Krieger Mind/Brain Institute, Johns Hopkins University, Baltimore, United States

**\*For correspondence:**
dan.oconnor@jhmi.edu

**Competing interest:** The authors declare that no competing interests exist.

**Abstract** Flexible responses to sensory stimuli based on changing rules are critical for adapting to a dynamic environment. However, it remains unclear how the brain encodes and uses rule information to guide behavior. Here, we made single-unit recordings while head-fixed mice performed a cross-modal sensory selection task where they switched between two rules: licking in response to tactile stimuli while rejecting visual stimuli, or vice versa. Along a cortical sensorimotor processing stream including the primary (S1) and secondary (S2) somatosensory areas, and the medial (MM) and anterolateral (ALM) motor areas, single-neuron activity distinguished between the two rules both prior to and in response to the tactile stimulus. We hypothesized that neural populations in these areas would show rule-dependent preparatory states, which would shape the subsequent sensory processing and behavior. This hypothesis was supported for the motor cortical areas (MM and ALM) by findings that (1) the current task rule could be decoded from pre-stimulus population activity; (2) neural subspaces containing the population activity differed between the two rules; and (3) optogenetic disruption of pre-stimulus states impaired task performance. Our findings indicate that flexible action selection in response to sensory input can occur via configuration of preparatory states in the motor cortex.

## eLife assessment

This **important** work advances our understanding of how brains flexibly gate actions in different contexts, a topic of great interest to the broader field of systems neuroscience. Recording neural activity from several sensory and motor cortical areas along a sensorimotor pathway, the authors found that preparatory activity in motor cortical areas of the mouse depends on the context in which an action will be carried out, consistent with previous theoretical and experimental work. Furthermore, the authors provide causal evidence that these changes support flexible gating of actions. The carefully carried out experiments were analyzed using state-of-the-art methodology and provide **convincing** conclusions.

## Introduction

Abstract relationships between objects, events, and actions can be established by rules. How we receive, process, and respond to sensory signals is guided by our understanding of rules that apply to the current context. For example, a student picks up a vibrating phone while waiting for a phone interview, but not during a lecture. This rule-guided flexibility is essential to adapt in a dynamic environment (*Miller and Cohen, 2001*) and is at the core of many advanced cognitive functions such as economic decision-making and social interaction (*Koechlin, 2014*; *Mansouri et al., 2020*). Several lines of studies have shown that impairments in rule-based decision-making are linked to

neurodevelopmental conditions including autism spectrum disorder (*Gastgeb et al., 2012*; *Jones et al., 2013*) and schizophrenia (*Weinberger and Berman, 1996*; *Woodward et al., 2012*).

Flexible rule-guided behaviors require at least two processes from the brain. First, (1) to maintain and update rule representations; and (2) to apply the appropriate rule to correctly transform sensory signals into motor outputs. Higher-order brain areas in frontal and parietal cortices are thought to play an important role in process 1, i.e., in encoding abstract rules and guiding the sensorimotor transformation (*Miller and Cohen, 2001*; *Bari et al., 2019*; *Hattori and Komiyama, 2022*; *Kamigaki et al., 2009*; *Rodgers and DeWeese, 2014*; *Sakai, 2008*). It remains less clear how process #2 occurs, i.e., how rules are applied to sensorimotor pathways to govern the mapping between stimulus and response (*van den Brink et al., 2023*).

Rodent orofacial circuits provide well-defined sensorimotor pathways to study rule implementation. Recent studies have uncovered important cortical areas linking whisker input to licking output in goal-directed behavior (*Crochet et al., 2019*; *Esmaeili et al., 2021*; *Petersen, 2019*; *Svoboda and Li, 2018*). For example, findings in a delayed tactile detection task have shown that the whisker region of primary somatosensory cortex is required for sensory detection and the anterior lateral motor cortex (ALM) is critical for motor planning and execution of licking (*Esmaeili et al., 2021*; *Guo et al., 2014*). Additionally, the medial motor cortex (MM), which contains the whisker primary and secondary motor cortices (*Mao et al., 2011*; *Matyas et al., 2010*), demonstrates early selectivity of whisker-based tactile signals compared with ALM (*Chen et al., 2017*).

Activity in sensorimotor pathways can reflect both sensory input and the rules dictating the appropriate use of that input (*Rodgers and DeWeese, 2014*; *van den Brink et al., 2023*; *Condylis et al., 2020*; *Steinmetz et al., 2019*; *Zhang et al., 2014*). Analysis of neural populations may provide insight into how these sensory and contextual signals are integrated to govern behavior. Recent progress in the study of neural population dynamics has uncovered population-level computations for motor control (*Afshar et al., 2011*; *Churchland et al., 2006a*; *Churchland et al., 2012*; *Sauerbrei et al., 2020*), timing (*Buonomano and Maass, 2009*; *Remington et al., 2018*; *Wang et al., 2018*), and decision-making (*Finkelstein et al., 2021*; *Gao et al., 2018*; *Mante et al., 2013*; *Raposo et al., 2014*). The evolution of the neural population state is controlled by initial conditions, internal dynamics, and external inputs, and can allow computations to be carried out (*Vyas et al., 2020*). For example, preparatory activity in non-human primate motor and premotor areas has been proposed to initialize dynamical systems to generate appropriate arm movements (*Churchland et al., 2010*; *Shenoy et al., 2013*). In addition, thalamic input has been shown to drive cortical dynamics in the mouse motor cortex during reaching (*Sauerbrei et al., 2020*). In the primate dorsomedial frontal cortex, the speed of neural trajectories encoding the passage of time is adjusted by both initial conditions and thalamic inputs (*Remington et al., 2018*; *Wang et al., 2018*).

Here, we investigate how task rules affect activity in a sensorimotor pathway to govern the mapping between stimulus and response. We trained mice in a cross-modal sensory selection task that we recently developed (*Chevée et al., 2022*) and then recorded and analyzed population single-unit activity from a set of cortical areas along the pathway that transforms whisker sensory inputs into licking motor outputs (*Esmaeili et al., 2021*; *Guo et al., 2014*; *Chen et al., 2017*). We found that single-neuron and population activity before stimulus delivery reflected task rules in the whisker regions of primary (S1) and secondary (S2) somatosensory cortical areas, and in the MM and ALM areas. Across these cortical areas, neural subspaces containing the trial activity differed between the two rules. Pre-stimulus population states in the motor cortical areas shifted in a manner that tracked the rule switch. Optogenetic inhibition designed to disrupt pre-stimulus states in the motor cortical areas impaired rule-dependent tactile detection. Together, our results show that the application of task rules—to link a stimulus to the correct response—involves rule-dependent configuration of pre-stimulus preparatory states within the sensorimotor cortex.

## Results
### Task rules modulate touch-evoked activity of individual neurons in the tactile processing stream

We trained head-fixed mice to perform a cross-modal sensory selection task (*Chevée et al., 2022*) in which they switched between 'respond-to-touch' and 'respond-to-light' rules in different blocks

of trials (*Figure 1a and b*). During the respond-to-touch rule, mice responded to a tactile stimulus by licking to the right reward port, and to withhold licking following a visual stimulus. During the respond-to-light rule, mice responded to a visual stimulus by licking to the left reward port, and withheld licking following a tactile stimulus. For each trial, the stimulus duration was 0.15 s and an answer period extended from 0.1 to 2 s from stimulus onset. Blocks of trials under the two rules alternated multiple times in a session (*Figure 1b*; 4–6 blocks per session, ~60 trials per block). No cue was immediately provided after rule switching, so mice detected the rule change through reward availability over the first few trials. On the 9th trial, a drop of water from the correct reward port was released following the stimulus to ensure switching (*Figure 1b*, black dots).

The two stimulus-response rules determined the behavioral relevance of sensory stimuli. For example, tactile stimuli were behaviorally relevant in respond-to-touch blocks but irrelevant in respond-to-light blocks. Four trial outcomes were defined based on the behavioral relevance of a sensory stimulus and on the response of the mouse. Correct licking responses following a relevant stimulus were 'hits', and correct withholding of responses following an irrelevant stimulus were 'correct rejections'. Failed responses to a relevant stimulus were 'misses', and incorrect licking responses (following an irrelevant stimulus and/or at the incorrect port) were 'false alarms'. Two sensory modalities and four trial outcomes composed eight trial types in the cross-modal sensory selection task (*Figure 1c*). Only hit trials were rewarded with a drop of water, and the other trial types were neither rewarded nor punished. We used stimulus strengths (tactile stimulus: single whisker, 20 Hz, 150 ms, ~800 degrees/s; visual stimulus: 470 nm LED, 150 ms, ~3 µW) that yielded performance of approximately 75% correct for both respond-to-touch and respond-to-light blocks (*Figure 1d*; touch: 74 ± 1%, light: 75 ± 1% [mean ± s.e.m.], p=0.56, paired sample t-test, *n*=12 mice). Thus, mice flexibly responded to tactile and visual stimuli in a rule-dependent manner.

To examine how the task rules influenced the sensorimotor transformation occurring in the tactile processing stream, we performed single-unit recordings from sensory and motor cortical areas including S1, S2, MM, and ALM (*Figure 1e–g*, *Figure 1—figure supplement 1a–h*, and *Figure 1—figure supplement 2a*; S1: 6 mice, 10 sessions, 177 neurons, S2: 5 mice, 8 sessions, 162 neurons, MM: 7 mice, 9 sessions, 140 neurons, ALM: 8 mice, 13 sessions, 256 neurons). We recorded from one of these cortical areas per session, using 64-channel silicon probes. Across neurons, activity varied by orders of magnitude (*Figure 1—figure supplement 2b*) and showed diverse selectivity. In tactile stimulus trials, some neurons in sensory and motor cortical areas showed prominent touch-evoked responses in both block types (e.g. *Figure 1—figure supplement 1c*). Most of these touch-evoked responses appeared enhanced and sustained at time points prior to licking in the respond-to-touch (tactile hit ['tHit']) blocks when compared with similar time points in the respond-to-light (tactile correct rejection ['tCR']) blocks (*Figure 1f* and *Figure 1—figure supplement 1a and e*). In contrast, some neurons only showed increased activity when mice made lick responses, regardless of the block type (*Figure 1—figure supplement 1d, f, and h*). Neurons could also show a mixture of these two response types (*Figure 1—figure supplement 1b and g*).

We first determined to what extent the rules modulated touch-evoked activity by comparing tactile correct trials between respond-to-touch and respond-to-light blocks (tHit vs tCR). We restricted analysis to the 150 ms period of stimulus delivery, to focus on rule-dependent processing that was not influenced by overt movements (97% of lick onsets occurred >150 ms after stimulus onset). We used ideal observer analysis to determine how well trial-by-trial activity of a single neuron could discriminate between the task rules (Materials and methods). We found that 18–33% of neurons in these cortical areas had area under the receiver-operating curve (ROC) (AUC) values significantly different from 0.5, and therefore discriminated between tHit and tCR trials (*Figure 1h*; S1: 28.8%, 177 neurons; S2: 17.9%, 162 neurons; MM: 32.9%, 140 neurons; ALM: 23.4%, 256 neurons; criterion to be considered significant: Bonferroni corrected 95% CI on AUC did not include 0.5 for at least three consecutive 10 ms time bins). Moreover, the distribution of AUC values for discriminative neurons showed that the majority had enhanced responses (AUC >0.5) to tactile stimuli that were behaviorally relevant. Therefore, touch-evoked responses were overall enhanced by the relevance of tactile stimuli according to the current stimulus-response rule.

## Pre-stimulus activity of single neurons signals task rules

In our cross-modal selection task, rules were block-based and there were no cues to indicate the current rule (except for the rule transition cue on the 9th trial of each block), so the mice were required

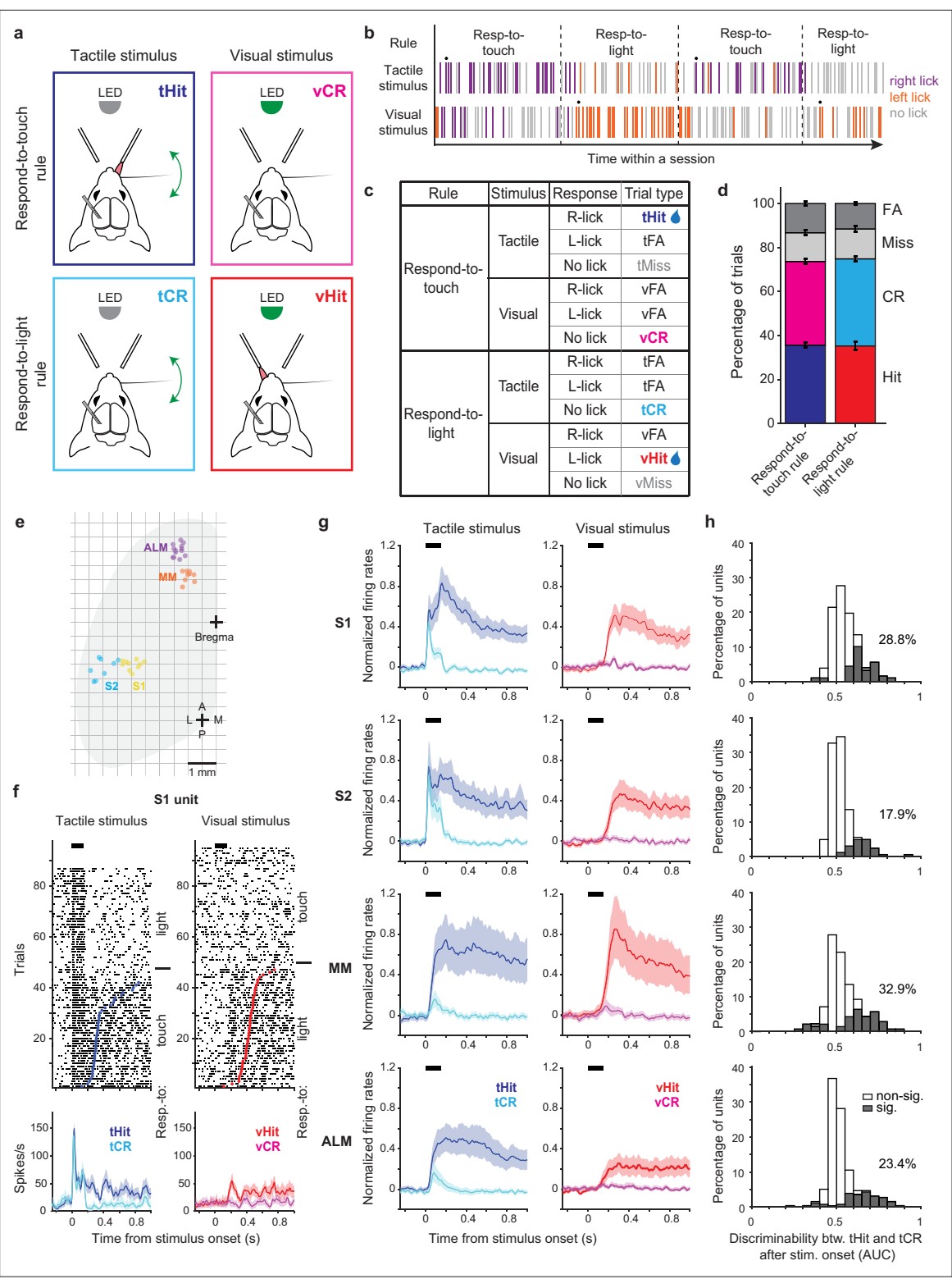

**Figure 1.** Touch-evoked activity of individual neurons is modulated by task rules. (**a**) Schematic of the cross-modal sensory selection task. Tactile and visual stimuli were associated with water reward during the respond-to-touch and respond-to-light rules, respectively. Mice were trained to lick to the right reward port after a tactile stimulus during the respond-to-touch rule, and lick to the left reward port after a visual stimulus during the respond-to-light rule. Mice withheld licking after a reward irrelevant stimulus (tactile stimulus during the respond-to-light rule or visual stimulus during the respond-to-touch rule). (**b**) Example behavioral session. Tactile and visual trials were randomly interleaved. Respond-to-touch and respond-to-light rules alternated in different blocks of trials during a behavioral session (~60 trials/block). The mouse adaptively changed its stimulus-response strategies

*Figure 1 continued on next page*

*Figure 1 continued*

based on the task rules. Each bar represents a trial and colors indicate lick responses (right lick: purple; left lick: orange; no lick: gray). A drop of water was delivered to the new reward port on the 9th trial (black dot) to cue mice to the rule switch. (**c**) Task design and trial outcomes. Correct licking responses were hits, and correct withholding of responses were correct rejections (CR). Failed responses were misses, and incorrect licking responses were false alarms (FA). Two sensory modalities and four types of trial outcomes comprise eight trial types. (**d**) The fractions of trial outcomes were similar between the respond-to-touch and respond-to-light rules (touch vs light: hit [36 ± 1% vs 35 ± 2%]; correct rejection [38 ± 1% vs 40 ± 1%]; miss [13 ± 1% vs 14 ± 1%]; false alarm [13 ± 1% vs 12 ± 1%]). The behavioral performance was ~75% correct for both rules (touch: 74 ± 1%, light: 75 ± 1%). Means ± s.e.m.; *n*=12 mice. (**e**) Reconstructed locations of silicon probes. S1: primary somatosensory cortex (6 mice, 10 sessions). S2: secondary somatosensory cortex (5 mice, 8 sessions). MM: medial motor cortex (7 mice, 9 sessions). ALM: anterolateral motor cortex (8 mice, 13 sessions). (**f**) Raster plots (top) and trial-averaged activity (bottom) of an example S1 unit. Correct tactile (left) and visual (right) trials were sorted by rule and response (tactile hit [tHit], blue; tactile correct rejection [tCR], cyan; visual hit [vHit], red; visual correct rejection [vCR], magenta). Dots indicate the first lick in hit trials. Thick black bars show the stimulus delivery window. Error shading: bootstrap 95% confidence interval (CI). (**g**) Normalized activity (z-score) across the population of recorded neurons in S1 (177 neurons), S2 (162 neurons), MM (140 neurons), and ALM (256 neurons). Error shading: bootstrap 95% CI. (h) Distribution of tHit and tCR discriminability for individual neurons. Discriminability of tHits and tCRs was defined as the ability of an ideal observer to discriminate tHits from tCRs on a trial-by-trial basis (0–150 ms; 10 ms bins). Approximately 15–35% of neurons showed significant difference between tHit and tCR responses (gray area; Bonferroni corrected 95% CI of area under the receiver-operating curve (AUC) does not include 0.5 for at least three consecutive time bins) in S1 (28.8%), S2 (17.9%), MM (32.9%), and ALM (23.4%).

The online version of this article includes the following figure supplement(s) for figure 1:

**Figure supplement 1.** Additional examples of single-unit responses during the cross-modal selection task.

**Figure supplement 2.** Mean firing rates across neurons and distributions of peak touch-evoked responses.

to maintain rule information during the intertrial interval (ITI). To test if ITI activity of neurons in the somatosensory and motor cortical areas reflected task rules, we first analyzed neural activity during the 1 s window preceding stimulus delivery. Trials with licking in this time window were removed to minimize possible movement effects on pre-stimulus activity (Material and methods). We found that some cortical neurons showed obvious changes in their pre-stimulus activity across blocks (*Figure 2a*). A preference for the respond-to-touch rule (i.e. with activity higher in respond-to-touch blocks compared with respond-to-light blocks) and a preference for the respond-to-light rule were both observed (*Figure 2a and b*). We next calculated discriminability between block types for each neuron to see how well an ideal observer could categorize the current trial's task rule on the basis of pre-stimulus activity (−100 to 0 ms from stimulus onset). To ensure mice were in the correct state to treat the following stimuli according to the rule, only correct tactile and visual trials were included. Less than 5% of neurons in S1 and S2 showed significant rule discriminability, while MM and ALM had around 10–20% significant neurons (*Figure 2c*; S1: 4.5%, 177 neurons; S2: 2.5%, 162 neurons; MM: 21.4%, 140 neurons; ALM: 10.2%, 256 neurons; Bonferroni corrected 95% CI on AUC did not include 0.5).

We also calculated the ability of each neuron to discriminate between tactile vs visual stimuli in windows before (−100 to 0 ms) or after (0–100 ms) stimulus onset, to serve as negative and positive controls, respectively, for our use of ideal observer analysis. As expected, single-unit activity before stimulus onset did not discriminate between tactile and visual trials (*Figure 2d*; S1: 0%, 177 neurons; S2: 0%, 162 neurons; MM: 0%, 140 neurons; ALM: 0.8%, 256 neurons). After stimulus onset, more than 35% of neurons in the sensory cortical areas and approximately 15% of neurons in the motor cortical areas showed significant stimulus discriminability (*Figure 2e*; S1: 37.3%, 177 neurons; S2: 35.2%, 162 neurons; MM: 15%, 140 neurons; ALM: 14.1%, 256 neurons).

The rule dependence of pre-stimulus activity indicates that neurons were in different states immediately prior to stimulus onset, suggesting that their responses to the subsequent sensory input might also differ. We therefore next investigated early touch-evoked responses (0–50 ms from stimulus onset) in those neurons that showed significant rule discriminability prior to stimulus delivery. We found that neurons with stronger preference for the respond-to-touch rule before stimulus onset showed larger touch-evoked activity in the respond-to-touch blocks, whereas neurons with a preference for the respond-to-light rule showed larger touch-evoked activity in the respond-to-light blocks (*Figure 2f*; Pearson correlation; S1: *r*=0.69, p=0.056, 8 neurons; S2: *r*=0.91, p=0.093, 4 neurons; MM: *r*=0.93, p<0.001, 30 neurons; ALM: *r*=0.83, p<0.001, 26 neurons). This result suggests that configuration of pre-stimulus states could play a role in achieving rule dependence of sensory processing.

Together, these results demonstrate that the pre-stimulus activity of single units in the sensory and motor cortical areas reflected task rules, with better discrimination of rules by units in the motor areas.

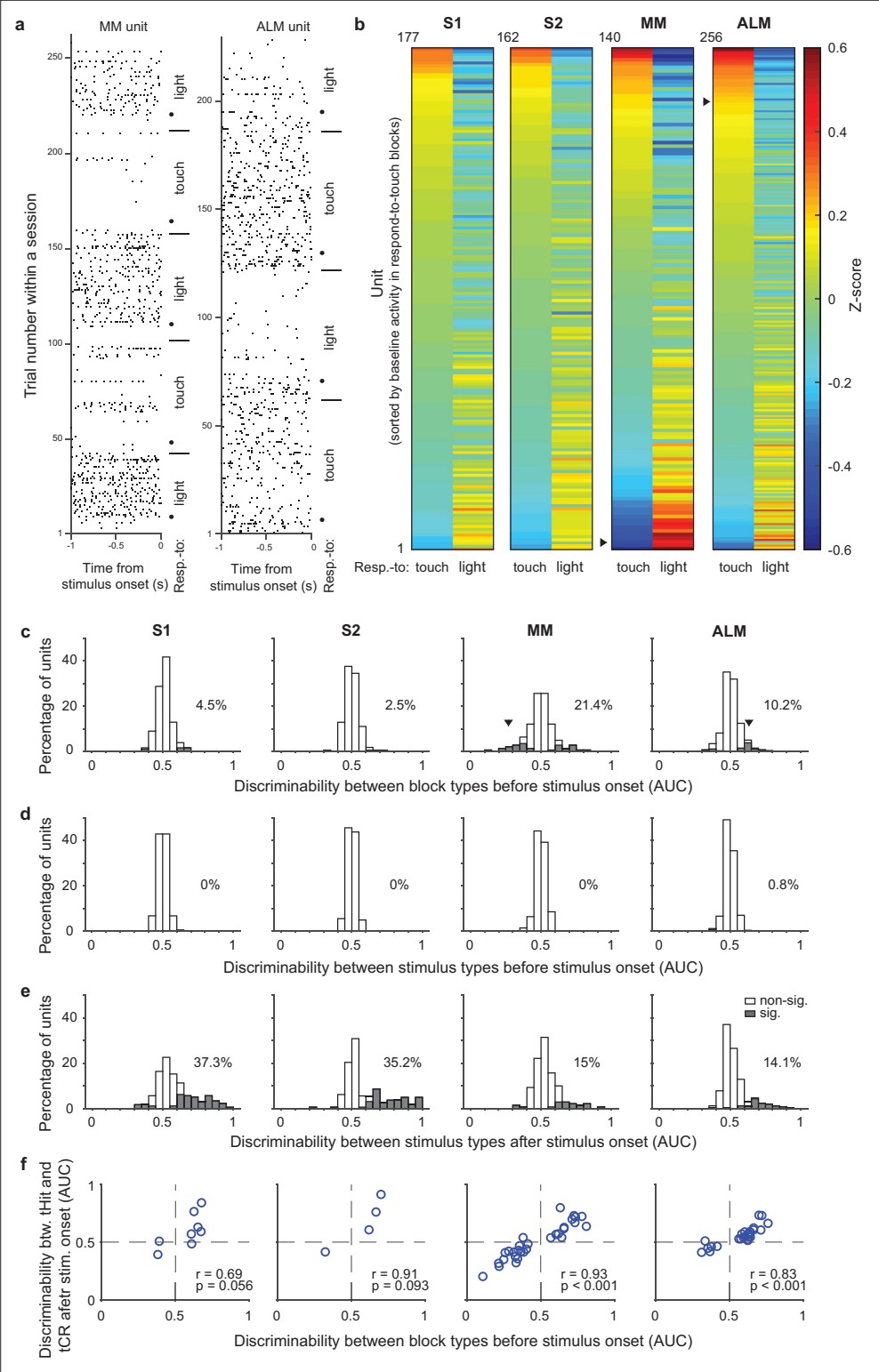

**Figure 2.** Single-unit activity before stimulus delivery reflects task rules. (**a**) Raster plots for example units from medial motor cortex (MM) (left) and anterolateral motor cortex (ALM) (right). Neural activity before stimulus delivery (−1 to 0 s from stimulus onset) changed across blocks within a session. Horizontal lines show block transitions and black dots indicate switch cues. (**b**) Heatmap of normalized (z-scored) pre-stimulus activity (−100 to 0 ms) across correct trials in respond-to-touch and respond-to-light blocks for units in S1 (*n*=177), S2 (*n*=162), MM (*n*=140), and ALM (*n*=256). Right-pointing triangles label example units in (**a**). (**c**) Distribution of respond-to-

*Figure 2 continued on next page*

*Figure 2 continued*

touch and respond-to-light discriminability for individual neurons. The respond-to-touch and respond-to-light discriminability was measured by the ability of an ideal observer to discriminate respond-to-touch from respond-to-light pre-stimulus activity (−100 to 0 ms) on a trial-by-trial basis (area under the receiver-operating curve [AUC]). The percentages of units showing significant discriminability (gray areas: Bonferroni corrected 95% CI of AUC did not include 0.5) were higher in motor cortical regions (MM: 21.4%; ALM: 10.2%) than in sensory cortical regions (S1: 4.5%; S2: 2.5%). Downward-pointing triangles label example units in (**a**). (**d**) Distribution of stimulus discriminability (tactile vs visual stimuli) for individual neurons during the pre-stimulus-onset window (−100 to 0 ms). Almost no units show significant stimulus discriminability before the stimulus onset (S1: 0%; S2: 0%; MM: 0%; ALM: 0.8%). (**e**) Same as (**d**) but for the post-stimulus-onset window (0–100 ms). A large percentage of units show significant stimulus discriminability after the stimulus onset (S1: 37.3%, S2: 35.2%, MM 15%, and ALM 14.1%). (**f**) Relationship between block-type discriminability before stimulus onset and tHit-tCR discriminability after stimulus onset for units showing significant block-type discriminability prior to the stimulus. Pearson correlation: S1: $r=0.69$, $p=0.056$, 8 neurons; S2: $r=0.91$, $p=0.093$, 4 neurons; MM: $r=0.93$, $p<0.001$, 30 neurons; ALM: $r=0.83$, $p<0.001$, 26 neurons. tHit, tactile hit; tCR, tactile correct rejection.

## Pre-stimulus states of neuronal populations reflect task rules

We next investigated the function of rule-dependent pre-stimulus activity from the perspective of neural population dynamics. In a dynamical system, state variables change based on their current values. This implies that the evolution of neural population activity during a trial will depend in part on the activity state of the population at the beginning of the trial (***Vyas et al., 2020***; ***Ebitz and Hayden, 2021***). We hypothesized that pre-stimulus activity—i.e., the state of the population at the start of the trial—would be set in such a way as to enable processing of the upcoming sensory signals according to the appropriate rule. Predictions from this hypothesis are that: (1) the pre-stimulus state of a neural population could be used to decode the current task rule; (2) the rule-dependent separation of neural subspaces before and after the tactile stimulus onset should be correlated; (3) pre-stimulus states would shift when the mice switched between rules; and (4) perturbations of the pre-stimulus state should disrupt task performance. We tested each prediction via the following analyses and experiments.

We first determined whether the pre-stimulus population state could be used to decode the current task rule. For each session, we used linear discriminant analysis (LDA) to obtain a classification accuracy for block type (respond-to-touch vs respond-to-light) based on the pre-stimulus activity (−100 to 0 ms relative to stimulus onset) of simultaneously recorded neurons on correct trials (***Figure 3a***). Pre-stimulus states in S1, S2, MM, and ALM could each be used to decode the block type (***Figure 3b***; medians of classification accuracy [true vs shuffled]: S1 [0.61 vs 0.5], 10 sessions; S2 [0.62 vs 0.53], 8 sessions; MM [0.7 vs 0.52], 9 sessions; ALM [0.68 vs 0.55], 13 sessions). Support vector machine (SVM) and Random Forests classifiers showed similar decoding abilities (***Figure 3—figure supplement 1a and b***; medians of classification accuracy [true vs shuffled]; SVM: S1 [0.6 vs 0.53], 10 sessions, S2 [0.61 vs 0.51], 8 sessions, MM [0.71 vs 0.51], 9 sessions, ALM [0.65 vs 0.52], 13 sessions; Random Forests: S1 [0.59 vs 0.52], 10 sessions, S2 [0.6 vs 0.52], 8 sessions, MM [0.65 vs 0.49], 9 sessions, ALM [0.7 vs 0.5], 13 sessions).

Interestingly, activity in the motor cortical areas allowed robust block-type classification (no overlap between bootstrap 95% CIs for the true and shuffled data, 95% CIs for true vs shuffled data: MM [0.60,0.73] vs [0.48, 0.53]; ALM [0.56,0.66] vs [0.49, 0.54]). In contrast, the sensory cortical areas showed limited block-type discriminability (***Figure 3c***; bootstrap 95% CIs for the true data were above 0.5 but overlapped with the bootstrap 95% CIs for the shuffled data, 95% CIs for true vs shuffled data: S1 [0.52,0.61] vs [0.49,0.55]; S2 [0.53,0.63] vs [0.49, 0.54]). In positive and negative control analyses, we found that neural population activity could be used to discriminate between stimulus types (tactile vs visual) after stimulus onset (***Figure 3e*** and ***Figure 3—figure supplement 1d***; 0–100 ms relative to stimulus onset) but not before stimulus onset (***Figure 3d*** and ***Figure 3—figure supplement 1c***; from −100 to 0 ms).

Together these results show that each of the two task rules was associated with a different pattern of pre-stimulus population activity in sensory and motor cortical areas, with a larger difference in the motor areas.

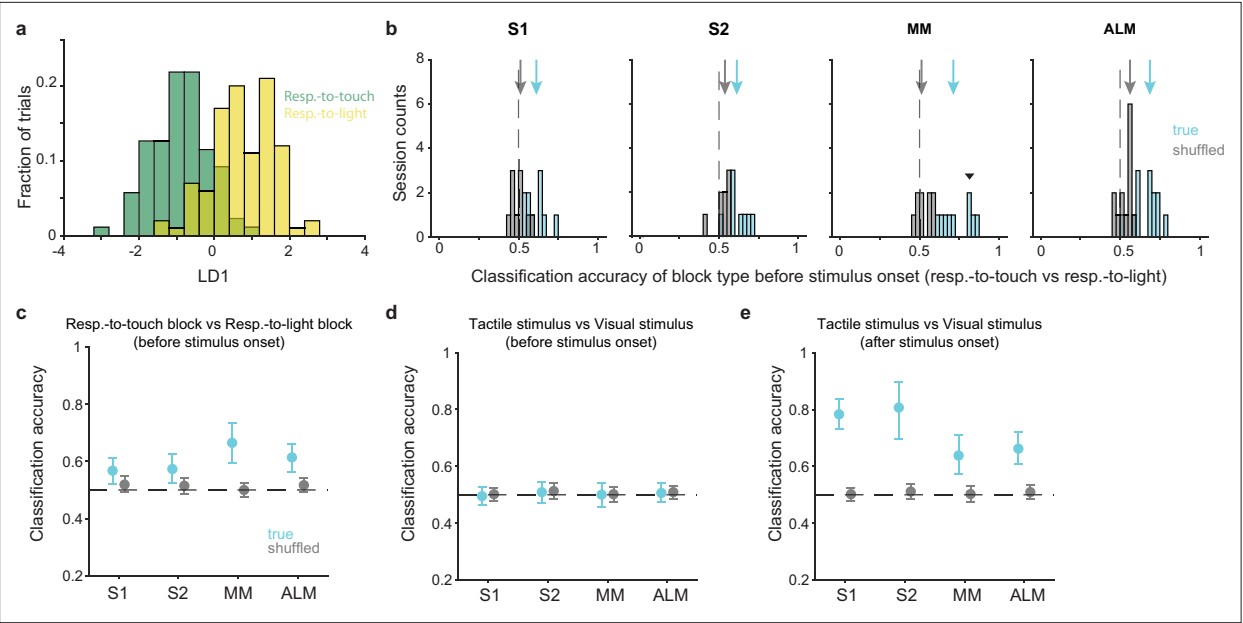

**Figure 3.** Pre-stimulus states of neural populations are rule-dependent. (**a**) Decoding of task rule (respond-to-touch vs respond-to-light) through linear discriminant analysis (LDA) for an example session. Pre-stimulus population activity (medial motor cortex [MM]; 21 neurons; –100 to 0 ms from stimulus onset) in respond-to-touch (green) and respond-to-light (yellow) blocks were projected onto the first linear discriminant (LD1). This plot shows the histogram of the projections onto the LD1 axis. (**b**) Distribution of classification accuracy for task rule (respond-to-touch vs respond-to-light) based on pre-stimulus activity for simultaneously recorded neurons in each session (S1: 10 sessions; S2: 8 sessions; MM: 9 sessions; anterior lateral motor cortex [ALM]: 13 sessions). The true (cyan) data showed better classification accuracy compared with the shuffled (gray) data (medians of classification accuracy [true vs shuffled]: S1 [0.61 vs 0.5]; S2 [0.62 vs 0.53]; MM [0.7 vs 0.52]; ALM [0.68 vs 0.55]). Arrows show classification accuracy medians. Dashed lines indicate the chance level (0.5). The downward-pointing triangle shows the example session in (**a**). (**c**) Session-averaged classification accuracy for task rules based on pre-stimulus population activity in S1 (95% CI of true [cyan] and shuffled [gray] data: true [0.52,0.61], shuffled [0.49,0.55]), S2 (true [0.53,0.63], shuffled [0.49, 0.54]), MM (true [0.60,0.73], shuffled [0.48, 0.53]) and ALM (true [0.56,0.66], shuffled [0.49, 0.54]). Error bars show bootstrap 95% CI. (**d**) Same as (**c**) but for the classification accuracy for stimulus types. (**e**) Same as (**c**) but for the classification accuracy for stimulus types based on population activity after stimulus onset (0–100 ms).

The online version of this article includes the following figure supplement(s) for figure 3:

**Figure supplement 1.** Classification accuracy for task rules and stimulus types.

## Separation of pre-stimulus states predicts subsequent divergent processing

We found pre-stimulus population activity was rule-dependent across sensory and motor cortical areas. We next asked if these rule-dependent pre-stimulus states affect post-stimulus neural activity. We investigated the relationship between the difference in pre-stimulus states between tHits and tCRs and the divergence of their subsequent neural trajectories. To assess this for the four cortical areas, we quantified how the tHit and tCR trajectories diverged from each other by calculating the Euclidean distance between matching time points for all possible pairs of tHit and tCR trajectories for a given session and then averaging these for the session (***Figure 4a and b***; S1: 10 sessions, S2: 8 sessions, MM: 9 sessions, ALM: 13 sessions, individual sessions in gray and averages across sessions in black; window of analysis: –100 to 150 ms relative to stimulus onset; 10 ms bins; using the top three principal components [PCs]; Materials and methods). The resulting time series of distance values from all sessions (*n*=40) were then ranked according to their means over the 100 ms period preceding stimulus onset and split into two groups, those above and below the median. The top 50% group showed a larger mean distance between tHit and tCR trajectories after stimulus onset compared with the bottom 50% group (***Foffani and Moxon, 2004***; ***O'Connor et al., 2010***; ***Figure 4c***; permutation test, p<0.001, 40 sessions). This result shows that the difference in population responses to the tactile stimuli under the two rules is commensurate with the difference in pre-stimulus states.

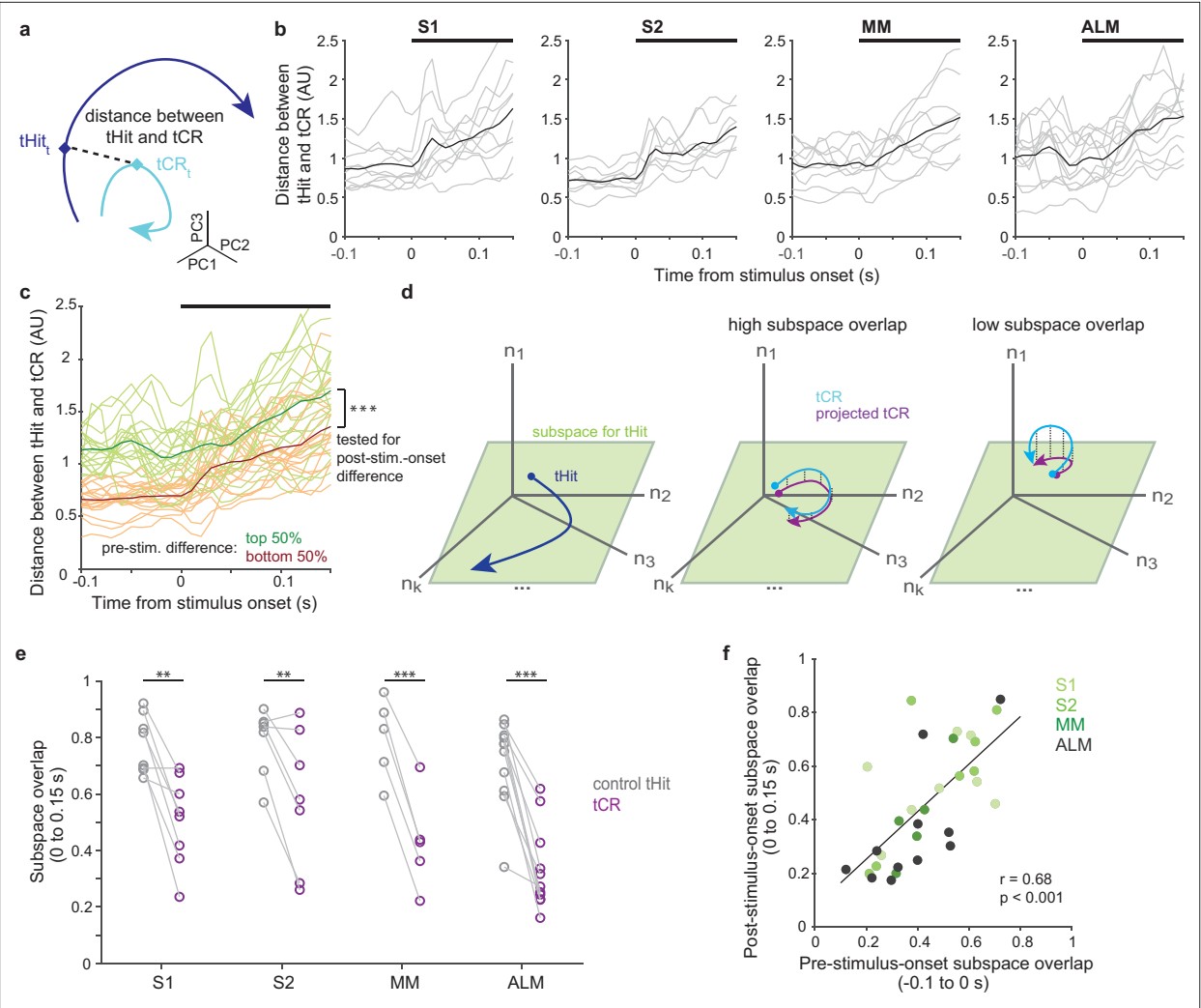

**Figure 4.** Pre-stimulus states predict subsequent tactile processing. (**a**) Schematic of distance (dashed line) between tactile hit (tHit) (blue) and tactile correct rejection (tCR) (cyan) trajectories. (**b**) Distance between tHit and tCR trajectories in S1, S2, medial motor cortex (MM), and anterior lateral motor cortex (ALM). Gray traces show the time varying tHit-tCR distance in individual sessions and black traces are session-averaged tHit-tCR distance (S1:10 sessions; S2: 8 sessions; MM: 9 sessions; ALM: 13 sessions). (**c**) Distance between tHit and tCR trajectories. Traces show individual sessions (*n*=40) pooled across areas and were sorted and labeled based on the distance prior to the stimulus (top 50%: green; bottom 50%: orange). The post-stimulus-onset distance was larger in the top 50% group than the bottom 50% group (permutation test, p<0.001). In (**b**) and (**c**) thick black bars show periods of stimulus delivery. (**d**) Schematic of the subspace overlap between tHit and tCR. A subspace (green parallelogram) for population activity of tHit (blue) is in a high-dimensional neural state space (left panel). If the subspaces of tHit and tCR are aligned, the tHit subspace could explain much of the variance of tCR (high subspace overlap; middle panel). That is, the projection of tCR onto the tHit subspace (purple) preserves much of the variance of tCR (cyan). If the subspaces of tHit and tCR are unaligned, the projection of tCR onto the tHit subspace preserves little of the variance of tCR (right panel). (**e**) Subspace overlap for control tHit (gray) and tCR (purple) trials in the somatosensory and motor cortical areas. Each circle is a subspace overlap of a session. Paired t-test, tCR – control tHit: S1: –0.23, 8 sessions, p=0.0016; S2: –0.23, 7 sessions, p=0.0086; MM: –0.36, 5 sessions, p<0.001; ALM: –0.35, 11 sessions, p<0.001; significance: ** for p<0.01, *** for p<0.001. (**f**) Relationship between pre- and post-stimulus-onset subspace overlaps. The subspace overlaps between tHit and tCR trials, calculated during pre- and post-stimulus-onset periods, were correlated (Pearson correlation, *r*=0.68, p<0.001; linear regression [black line]; 31 sessions).

The online version of this article includes the following figure supplement(s) for figure 4:

**Figure supplement 1.** Stimulus and choice coding dimensions (CDs) in respond-to-touch and respond-to-light blocks.

## Neural subspaces for tactile processing are rule-dependent

Although the full dimensionality of a neural state space is equal to the number of neurons, correlations among neurons typically cause dynamics to occur within a subspace of lower dimensionality (*Ebitz and Hayden, 2021*; *Cunningham and Yu, 2014*). Population activity associated with each task rule

might not only follow distinct trajectories, but could also occur within different neural subspaces. To test this, we calculated the overlap between the subspaces associated with tHit and tCR trials (*Raposo et al., 2014*; *Elsayed et al., 2016*; *Russo et al., 2020*; *Figure 4d*; see Materials and methods). We expect that, if tHit and tCR trial activity occupies largely overlapping subspaces, then the neural dimensions capturing most of the tHit activity will also explain much of the tCR activity (*Figure 4d*, middle). Conversely, if they occupy largely distinct subspaces, then the dimensions capturing most of the tHit activity will explain little of the tCR activity (*Figure 4d*, right). For each session, tHit trials were divided randomly into equally sized 'reference' and 'control' groups. The reference tHit trials were then used to perform a PC analysis (PCA; using data 0–150 ms from stimulus onset; 10 ms bins). We projected activity from tCR trials and from the control tHit trials into the space of the top three PCs obtained from the reference group PCA, then calculated and normalized their variance explained (Materials and methods). For S1, S2, MM, and ALM, subspace overlaps for tCR trials were significantly lower than the corresponding subspace overlaps for the control tHit trials (*Figure 4e*, purple vs gray symbols; tCR – control tHit: S1 [–0.23], 8 sessions, p=0.0016; S2 [–0.23], 7 sessions, p=0.0086; MM [–0.36], 5 sessions, p =<0.001; ALM [–0.35], 11 sessions, p< 0.001, paired t-test). This finding suggests that different neural subspaces were used for processing tactile stimuli under each of the two rules, in both sensory and motor cortical areas.

We next asked if the rule-dependent separation of subspaces during stimulus delivery was related to how the subspaces were separated prior to the tactile stimulus. This would provide evidence that differences in tactile stimulus processing follow from differences in the state of the neural population at the time the stimulus is received. We found that, across the cortical areas, the subspace overlaps for tCR trials calculated from periods before and after the tactile stimulus onset were correlated (*Figure 4f*; Pearson correlation, *r*=0.68, p<0.001; 31 sessions). This indicates that the shift between neural subspaces associated with each rule occurred prior to stimulus delivery and should thus impact processing of the stimulus.

Together, our results suggest that cortical populations are 'pre-configured' according to the current rule, such that an incoming sensory signal leads to distinct processing and ultimately distinct actions.

## Choice coding dimensions change with task rule

Gating of sensory information involves changing how sensory information is represented and read out (*Finkelstein et al., 2021*). This can be achieved by shifting sensory and/or choice coding dimensions (CDs) in the population activity space (*Mante et al., 2013*; *Ruff and Cohen, 2019*). In the previous section, we showed that neural subspaces containing trial dynamics changed between the two task rules. We next asked whether stimulus and choice CDs within these subspaces also shifted with task rules. For each task rule, we estimated CDs that maximally discriminated the neural trajectories for different conditions (*Li et al., 2016*; *Yang et al., 2022*; *Figure 4—figure supplement 1a–d*; Materials and methods). Since mice rarely licked the wrong port for a given block (*Figure 1b*), right-lick and no-lick trials were used to obtain choice CDs for the respond-to-touch blocks, and left-lick and no-lick trials for the respond-to-light blocks. To assess whether stimulus and choice CDs changed with task rule, we calculated for each session the dot product between the CDs obtained from respond-to-touch and respond-to-light blocks. We then used the magnitude of this dot product as an unsigned measure of the relative orientations of the CDs. In ALM, the dot product magnitudes calculated between stimulus CDs for the two block types were not significantly different from those calculated after shuffling trial-type labels (*Figure 4—figure supplement 1e*; significance defined as non-overlap of 95% CIs). This suggests that the stimulus CD in a respond-to-touch block had an orientation unrelated to that in a respond-to-light block. In contrast, we found that S1, S2, and MM had stimulus CDs that were significantly aligned between the two block types (*Figure 4—figure supplement 1e*; magnitude of dot product between the respond-to-touch stimulus CDs and the respond-to-light stimulus CDs, mean ± 95% CI for true vs shuffled data: S1: 0.5 ± [0.34, 0.66] vs 0.21 ± [0.12, 0.34], 10 sessions; S2: 0.62 ± [0.43, 0.78] vs 0.22 ± [0.13, 0.31], 8 sessions; MM: 0.48 ± [0.38, 0.59] vs 0.24 ± [0.16, 0.33], 9 sessions; ALM: 0.33 ± [0.2, 0.47] vs 0.21 ± [0.13, 0.31], 13 sessions). In contrast, the choice CDs for the two block types were not aligned well in S1, S2, MM, or ALM (*Figure 4—figure supplement 1f*; magnitude of dot product between the respond-to-touch choice CD and the respond-to-light choice CD, mean ± 95% CI for true vs shuffled data: S1: 0.28 ± [0.15, 0.43] vs 0.21 ± [0.12, 0.33], 10 sessions; S2: 0.22 ± [0.11, 0.33] vs 0.21 ± [0.13, 0.32], 8 sessions; MM:

0.22 ± [0.13, 0.33] vs 0.22 ± [0.14, 0.3], 9 sessions; ALM: 0.27 ± [0.16, 0.39] vs 0.21 ± [0.13, 0.31], 13 sessions).

Choice CDs were obtained from right-lick and no-lick trials in respond-to-touch blocks and left-lick and no-lick trials in respond-to-light blocks. Because the required lick directions differed between the block types, the difference in choice CDs across task rules (*Figure 4—figure supplement 1f*) could have been affected by the different motor responses. To rule out this possibility, we did a new version of this analysis using right-lick and left-lick trials to calculate the choice CDs for both task rules. We found that the orientation of the choice CD in a respond-to-touch block was still not aligned well with that in a respond-to-light block (*Figure 4—figure supplement 1h*; magnitude of dot product between the respond-to-touch choice CD and the respond-to-light choice CD, mean ±95% CI for true vs shuffled data: S1: 0.39 ± [0.23, 0.55] vs 0.2 ± [0.1, 0.31], 10 sessions; S2: 0.32 ± [0.18, 0.46] vs 0.2 ± [0.11, 0.3], 8 sessions; MM: 0.35 ± [0.21, 0.48] vs 0.18 ± [0.11, 0.26], 9 sessions; ALM: 0.28 ± [0.17, 0.39] vs 0.21 ± [0.12, 0.31], 13 sessions).

Overall, these results suggest that the different subspaces for tactile processing under the two rules result at least in part from changes in choice CDs.

## Pre-stimulus states in motor cortex track behavioral rule switches

Task rules switched multiple (three to five) times in each behavioral session. Mice detected a rule switch either through trial and error during the first few trials after the switch, or when a drop of water from the correct reward port was given on the 9th trial (which served as a cue to ensure that mice switched by this point; *Figure 1b*). The probabilities of right-licks and left-licks showed that the mice switched their motor responses during block transitions depending on task rules (*Figure 5a*, mean ± 95% CI across 12 mice). We used the first hit trial as the mark of a successful behavioral switch and found that mice switched before or immediately after the cue (*Figure 5b*, total number of block switches: respond-to-touch: 88 switches, respond-to-light: 91 switches). To analyze how pre-stimulus states changed over the course of a rule transition, we defined a 'transition period' that spanned from the first trial after a block switch until the first hit trial of the new block. In addition, we divided each transition period into 'early' and 'late' parts based on the occurrence of the first false alarm trial of the new block (*Figure 5c*). We considered the first false alarm trial to be the point at which the mouse first received feedback to indicate that a block change had occurred.

We hypothesized that the behavioral change that marked a successful switch in rule application would be accompanied by a neural change. Specifically, that the pre-stimulus population state would progress from that typical of the prior type of block to that typical of the new type of block in parallel with the behavioral shift. To test this, we trained an LDA classifier to discriminate respond-to-touch block and respond-to-light block trials using pre-stimulus neural activity. We used 90% of the correct trials as training data and tested classifier performance on the held-out 10% of correct trials (*Figure 5d and e*). Using our 'transition period' definition as described above, we tested classifier performance on 'early' transition and 'late' transition trials (*Figure 5c*). For respond-to-touch to respond-to-light block transitions, the fractions of trials classified as respond-to-touch for MM and ALM decreased progressively over the course of the transition (*Figure 5d*; rank correlation of the fractions calculated for each of the separate periods spanning the transition, Kendall's tau, mean ±95% CI: MM: –0.39 ± [–0.67,–0.11], 9 sessions, ALM: –0.29 ± [–0.54,–0.04], 13 sessions; criterion to be considered significant: 95% CI on Kendall's tau did not include 0). Similarly, the fractions of trials classified as respond-to-touch increased progressively over the course of transitions from respond-to-light to respond-to-touch blocks (*Figure 5e*; Kendall's tau, mean ± 95% CI: MM: 0.37 ± [0.07, 0.63], 9 sessions, ALM: 0.27 ± [0.03, 0.49], 13 sessions). Accuracies for classification of trials by block type based on S1 and S2 activity were uniformly poor and showed no clear trends across block transitions (*Figure 5—figure supplement 1*; mean ± 95% CI on Kendall's tau for respond-to-touch → respond-to-light transitions: S1: –0.16 ± [–0.43, 0.1], 10 sessions, S2: –0.15 ± [–0.54, 0.21], 8 sessions; for respond-to-light → respond-to-touch transitions: S1: 0.21 ± [–0.07, 0.5], 10 sessions, S2: 0.25 ± [–0.17, 0.63], 8 sessions). Together, these results indicate that the pre-stimulus states of neural populations in MM and ALM shifted over the course of block transitions in a manner commensurate with the behavioral shift in rule application.

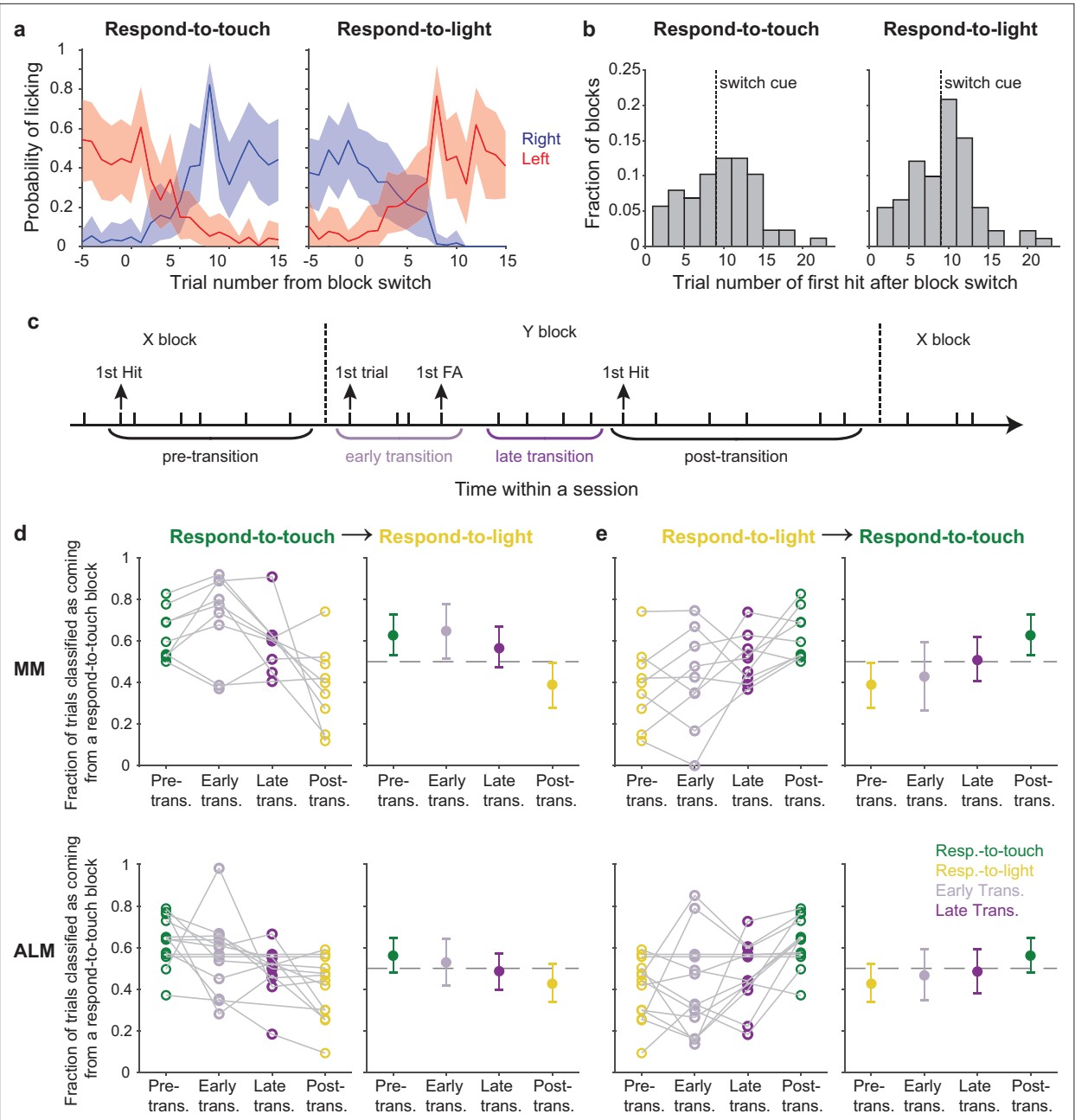

**Figure 5.** Pre-stimulus states of the motor cortical areas track behavioral rule switching. (**a**) Probabilities of right-licks (blue) and left-licks (red) during block transitions (left panel: respond-to-light → respond-to-touch transitions; right panel: respond-to-touch → respond-to-light transitions). Mean ± 95% CI across 12 mice. (**b**) Histogram showing the distribution of the trial number of the first hit after block switch. Most first hits in respond-to-touch (left) and respond-to-light (right) blocks occurred before or immediately after the switch cue (dashed line). (**c**) Schematic of rule transition in the cross-modal selection task. A transition period was defined as spanning from the first trial after the block switch until the first hit trial of that block. 'Early' and 'late' transitions were separated by the first false alarm trial of that block. (**d**) Fraction of trials classified as coming from a respond-to-touch block based on the pre-stimulus population state, for trials occurring in different periods (see **c**) relative to respond-to-touch → respond-to-light transitions. For medial motor cortex (MM) (top row) and anterior lateral motor cortex (ALM) (bottom row), progressively fewer trials were classified as coming from the respond-to-touch block as analysis windows shifted later relative to the rule transition. Kendall's tau (rank correlation): MM: –0.39, 9 sessions; ALM: –0.29, 13 sessions. Left panels: individual sessions, right panels: mean ± 95% CI. Dashed lines are chance levels (0.5). (**e**) Same as (**d**) but for respond-to-light → respond-to-touch transitions. Kendall's tau: MM: 0.37, 9 sessions; ALM: 0.27, 13 sessions.

The online version of this article includes the following figure supplement(s) for figure 5:

**Figure supplement 1.** Pre-stimulus states of neural populations in the sensory cortex during rule transitions.

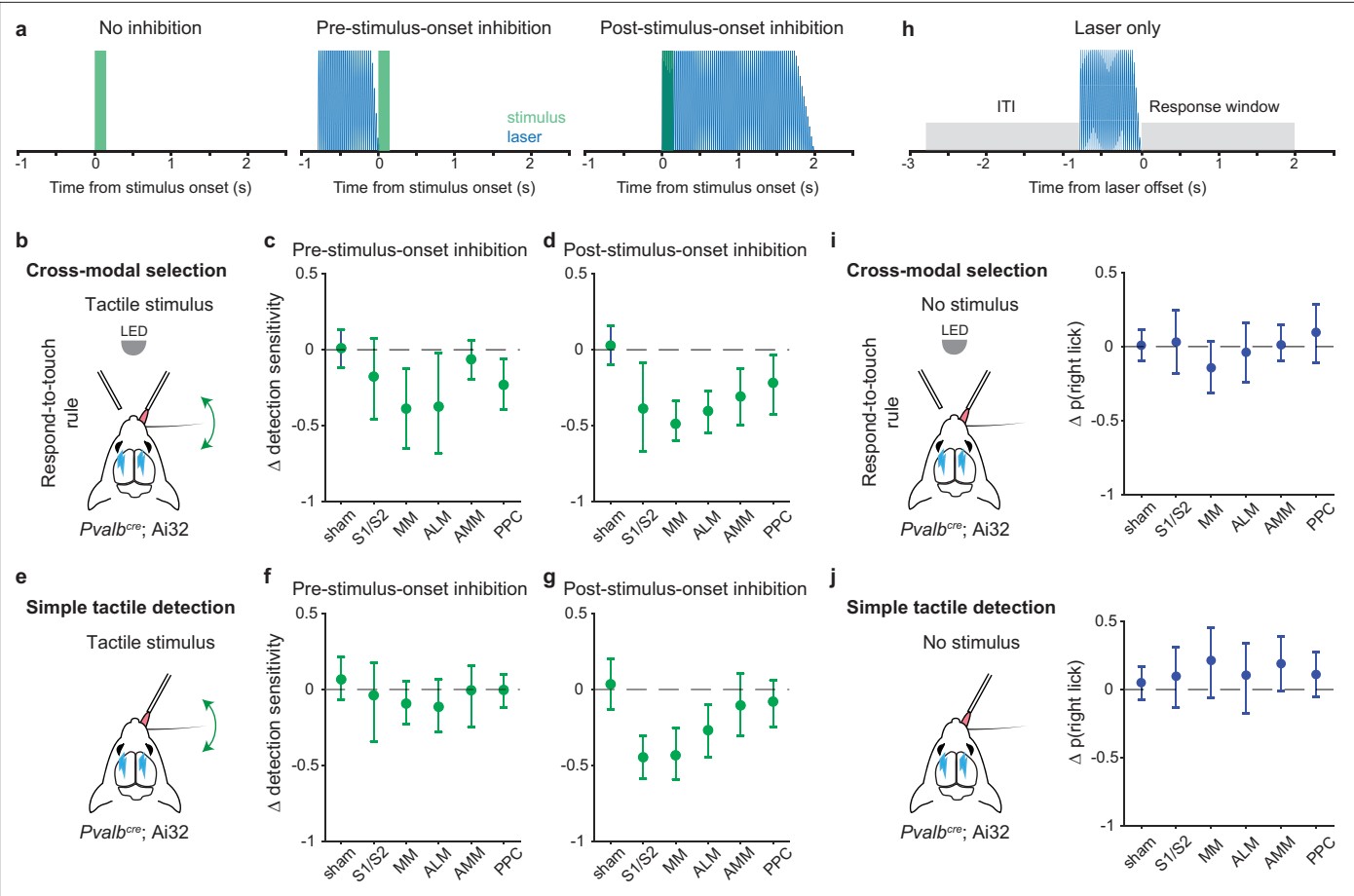

**Figure 6.** Inhibition of pre-stimulus states in the motor cortex impairs rule-dependent tactile detection. (**a**) Schematic of inhibition conditions for stimulus trials. Sessions comprised 80% stimulus trials and 20% laser-only trials (see **h**). In 15% of the stimulus trials, optogenetic inhibition occurred before tactile or visual stimuli (middle, –0.8 to 0 s from stimulus onset) to suppress pre-stimulus activity in the targeted cortical area. In another 15% of stimulus trials, optogenetic inhibition began simultaneously with the stimulus onset (right, 0–2 s from stimulus onset) in order to suppress sensory-evoked activity. (**b–d**) Changes in detection sensitivity for tactile stimuli during respond-to-touch blocks in the cross-modal selection task when each cortical area was optogenetically inhibited. For pre-stimulus-onset inhibition (**c**), detection sensitivity decreased when medial motor cortex (MM) and anterior lateral motor cortex (ALM) but not S1 and S2 were suppressed. For post-stimulus-onset inhibition (**d**), the detection sensitivity decreased when sensory and motor cortical regions were suppressed. S1/S2: 4 mice, 10 sessions; MM: 4 mice, 11 sessions; ALM: 4 mice, 10 sessions. Sham: 7 mice, 28 sessions. AMM: anteromedial motor cortex, 7 mice, 21 sessions. PPC: posterior parietal cortex, 7 mice, 21 sessions. (**e–g**) Same as (**b–d**) but for a simple tactile detection task (not cross-modal selection task). Detection sensitivity for tactile stimuli was reduced when the sensory (**S1 and S2**) and motor (MM and ALM) cortical areas were inhibited during the post-stimulus-onset window (**g**) but not during the pre-stimulus-onset window (**f**). Sham: 5 mice, 9 sessions; S1/S2: 3 mice, 3 sessions; MM: 3 mice, 5 sessions; ALM: 3 mice, 4 sessions; AMM: 3 mice, 4 sessions; PPC: 3 mice, 4 sessions. (**h**) Schematic of inhibition conditions for 'laser-only' trials. The probability of licking during the response window (0–2 s from laser offset) was compared with the probability of licking during the intertrial interval (ITI, –2 to 0 s from laser onset). (**i**) Changes in the probability of right licks within the laser-only trials during respond-to-touch blocks in the cross-modal selection task. (**j**) Changes in the probability of right licks within the laser-only trials in the simple tactile detection task. Error bars show bootstrap 95% CI. Criterion to be considered significant: 95% CI did not include 0.

The online version of this article includes the following figure supplement(s) for figure 6:

**Figure supplement 1.** Hit and false alarm rates during the inhibition of population states in the dorsal cortex.

**Figure supplement 2.** Inhibition of pre-stimulus states in the medial parts of motor cortex impairs rule-dependent visual detection.

## Disruption of pre-stimulus motor cortical state impairs rule-dependent tactile detection

So far, we have shown that pre-stimulus neural population states in the somatosensory and motor cortical areas differed between the two task rules. This suggests that pre-stimulus activity may play a critical role in our task. To test this, we bilaterally inhibited the different cortical areas shortly before stimulus onset via optogenetic activation of parvalbumin-positive (*Pvalb*) GABAergic neurons (***Guo***

*et al., 2014*; *Sachidhanandam et al., 2013*; *Figure 6a*, middle panel; –0.8 to 0 s from stimulus onset). Additionally, we included two other areas of the dorsal cortex, the anteromedial part of the motor cortex (AMM) and the posterior parietal cortex (PPC). We recorded from AMM in our cross-modal sensory selection task and observed visually evoked activity (*Figure 1—figure supplement 1i–k*), suggesting that AMM may play an important role in rule-dependent visual processing. PPC contributes to multisensory processing (*Mohan et al., 2018*; *Olcese et al., 2013*; *Song et al., 2017*) and sensory-motor integration (*Le Merre et al., 2018*; *Funamizu et al., 2016*; *Gallero-Salas et al., 2021*; *Goard et al., 2016*; *Harvey et al., 2012*; *Mohan et al., 2019*). Therefore, we wanted to test the roles of these areas in our cross-modal sensory selection task. Finally, we also performed negative control (sham) sessions that were identical except that the optogenetic light path was obstructed.

We defined detection sensitivity for tactile stimuli as the difference between tHit rate and visual false alarm rate during the response-to-touch blocks. Tactile detection sensitivity was significantly decreased when MM and ALM but not S1 and S2 were inhibited during the pre-stimulus period (*Figure 6c*; criterion to be considered significant: 95% CI on Δ tactile sensitivity did not include 0; S1/S2: [–0.47, 0.08], 4 mice, 10 sessions; MM: [–0.65,–0.12], 4 mice, 11 sessions; ALM: [–0.68,–0.02], 4 mice, 10 sessions). This was primarily due to a reduction in the tHit rate (*Figure 6—figure supplement 1b*; 95% CI on Δ tHit rate: S1/S2 [–0.31, 0.01], 4 mice, 10 sessions; MM [–0.49,–0.15], 4 mice, 11 sessions; ALM [–0.46,–0.07], 4 mice, 10 sessions). Inhibition of S1, S2, MM, and ALM during a 2 s window starting at the time of the stimulus onset served as a positive control (*Figure 6a*, right panel). Consistent with previous studies (*Guo et al., 2014*; *Le Merre et al., 2018*), inhibition of these cortical areas after stimulus onset reduced detection sensitivity for tactile stimuli (*Figure 6d*; 95% CI on Δ tactile sensitivity: S1/S2 [–0.67,–0.08], 4 mice, 10 sessions; MM [–0.6,–0.34], 4 mice, 11 sessions; ALM [–0.55,–0.27], 4 mice, 10 sessions). Neither the negative control (sham) condition nor inhibition of AMM before stimulus onset showed an effect on the detection sensitivity for tactile stimuli (*Figure 6c*; sham: [–0.12, 0.13], 7 mice, 28 sessions; AMM [–0.19, 0.06], 7 mice, 21 sessions). Together, our results suggest that the pre-stimulus network activity states in MM and ALM play an important role in rule-dependent tactile processing.

We defined detection sensitivity for visual stimuli as the difference between visual hit rate and tactile false alarm rate during the respond-to-light blocks. Visual detection sensitivity was not affected when S1 and S2 were inhibited before (*Figure 6—figure supplement 2g*; 95% CI on Δ visual sensitivity: [–0.27, 0.05]) or after (*Figure 6—figure supplement 2h*; [–0.44, 0.04], 4 mice, 10 sessions) the stimulus onset. For pre-stimulus-onset inhibition of the motor cortical areas, visual detection sensitivity was decreased when MM and AMM but not ALM were suppressed (*Figure 6—figure supplement 2g*; MM [–0.6,–0.18], 4 mice, 11 sessions; ALM [–0.5, 0.04], 4 mice, 10 sessions; AMM [–0.35,–0.08], 7 mice, 21 sessions). Inhibition of any of the three motor cortical areas after stimulus onset caused a reduction in visual sensitivity (*Figure 6—figure supplement 2h*; MM [–0.83,–0.52]; ALM [–0.84,–0.39]; AMM [–0.72,–0.36]). Together, these results indicate that pre-stimulus states of neural populations in the medial parts of motor cortex, such as MM and AMM, are critical for processing visual stimuli in a rule-dependent manner.

PPC is involved in multisensory processing (*Mohan et al., 2018*; *Olcese et al., 2013*; *Song et al., 2017*) and decision-making (*Funamizu et al., 2016*; *Gallero-Salas et al., 2021*; *Goard et al., 2016*; *Harvey et al., 2012*; *Mohan et al., 2019*). Here, we tested for a critical role of PPC in our cross-modal selection task (7 mice, 21 sessions total). Detection sensitivities for both tactile and visual stimuli were decreased when PPC was inhibited before (*Figure 6c*; Δ tactile sensitivity: [–0.4,–0.06]; *Figure 6—figure supplement 2g*; Δ visual sensitivity [–0.31,–0.05]) or after the stimulus onset (*Figure 6d*; Δ tactile sensitivity: [–0.43,–0.04]; *Figure 6—figure supplement 2h*; Δ visual sensitivity: [–0.54,–0.07]). These reductions in tactile and visual detection sensitivities were caused by a decrease in hit rate and/or an increase in false alarm rate. In general, inhibition of PPC before stimulus onset decreased the hit rate (*Figure 6—figure supplement 1b*; Δ tHit [–0.36,–0.05]; *Figure 6—figure supplement 2b*; Δ visual hit [–0.26, 0]), whereas inhibition of PPC after stimulus onset increased the false alarm rate (*Figure 6—figure supplement 1f*; Δ visual false alarm [0.13, 0.38]; *Figure 6—figure supplement 2f*; Δ tactile false alarm [0, 0.25]).

It is possible that disruption of pre-stimulus states may affect aspects of tactile sensory processing and/or lick production that are unrelated to rules. To exclude this possibility, in a new cohort of mice (*n*=5), we inhibited each cortical area in either the pre- or the post-stimulus-onset period during

performance of a simple tactile detection task (*Figure 6e–g*). In this task, mice had only to report with Go/NoGo licking whether a whisker was deflected, without rule-switching components to the task or the need to suppress responses to distracting stimuli. We found that tactile sensitivity was decreased when the sensory and motor cortical areas were inhibited after but not before stimulus onset (*Figure 6f and g*; 95% CI on Δ tactile sensitivity for pre-stimulus-onset inhibition: S1/S2 [–0.34, 0.18], 3 mice, 3 sessions; MM [–0.23, 0.06], 3 mice, 5 sessions; ALM [–0.28, 0.07], 3 mice, 4 sessions; Δ tactile sensitivity for post-stimulus-onset inhibition: S1/S2 [–0.59,–0.31]; MM [-0.59,–0.25]; ALM [–0.45,–0.1]). This was mainly caused by the decrease in hit rate in the post-stimulus-onset inhibition (*Figure 6—figure supplement 1i*; 95% CI on Δ tHit rate: S1/S2 [–0.61,–0.28]; MM [–0.67,–0.2]; ALM [–0.73,–0.05]). Inhibition of AMM and PPC did not influence tactile sensitivity in either inhibition condition (*Figure 6f and g*; Δ tactile sensitivity for pre-stimulus-onset inhibition: AMM [–0.25, 0.16], 3 mice, 4 sessions; PPC [–0.12, 0.1], 3 mice, 4 sessions; Δ tactile sensitivity for post-stimulus-onset inhibition: AMM [–0.17, 0.18]; PPC [–0.14, 0.3]). Together, these results show that inhibition immediately prior to stimulus delivery did not impact the performance of a simple tactile detection task in which the stimulus-response rule remained fixed.

We conducted an additional analysis to rule out the possibility that the behavioral effects of cortical inhibition we observed could be due simply to a deficit in lick production per se. Specifically, we compared the probability of licking in laser-only trials (catch trials where there was no sensory stimulus) with the probability of licking during ITI, for both the cross-modal selection task and the simple tactile detection task (*Figure 6h*). Lick probability was unaffected during S1, S2, MM, and ALM experiments for both tasks, indicating that the behavioral effects were not due to an inability to lick (*Figure 6i and j*; 95% CI on Δ lick probability for cross-modal selection task: S1/S2 [–0.18, 0.24], 4 mice, 10 sessions; MM [–0.31, 0.03], 4 mice, 11 sessions; ALM [–0.24, 0.16], 4 mice, 10 sessions; Δ lick probability for simple tactile detection task: S1/S2 [–0.13, 0.31], 3 mice, 3 sessions; MM [–0.06, 0.45], 3 mice, 5 sessions; ALM [–0.18, 0.34], 3 mice, 4 sessions).

Taken together, our results suggest that the pre-stimulus states of motor cortical networks play an important role in rule-dependent sensorimotor transformations.

## Discussion

Here, we investigated how rules modulate the transformation of tactile signals into actions across a set of key sensory-motor cortical areas comprising S1, S2, MM, and ALM. We found that neural activity prior to stimulus delivery reflected task rules at both the single-neuron and population levels in each area, but more prominently so in the motor cortical areas MM and ALM. Across the areas examined, each of the two task rules was associated with its own neural subspace for processing tactile signals. In ALM and MM, pre-stimulus population states shifted concomitantly with the behavioral signs of a rule switch. Optogenetic inhibition of motor cortical areas during the pre-stimulus period impaired tactile detection during the cross-modal selection task, but not during a simpler tactile detection task that did not require switching among rules. Together, our results suggest that the neural population states in motor cortical areas ALM and MM play an important role in transforming sensory stimuli into actions in a flexible, rule-dependent manner.

The responses to tactile stimuli were enhanced in S1, S2, MM, and ALM when they were behaviorally relevant according to the current rule (*Figure 1h*). In our task, relevance relates to reward, movement preparation, and movement, factors known to influence activity across many brain areas (*Chen et al., 2017*; *Steinmetz et al., 2019*; *Chubykin et al., 2013*; *Lacefield et al., 2019*; *Musall et al., 2019*; *Shuler and Bear, 2006*). We chose not to attempt to dissociate 'relevance' from these factors in our task design, given that they are linked in many natural scenarios (*Moore and Zirnsak, 2017*; *Rizzolatti et al., 1987*). Below, we address potential concerns and confounding effects associated with reward and movement.

First, neural responses on tHit and tactile false alarms were similar, despite the fact that hits but not false alarms were rewarded (proportions of neurons showing a significant difference in mean response between tHit and tactile false alarms: S1 (0/177); S2 (2/162); MM (0/140); ALM (6/256); permutation tests on PSTHs with Bonferroni correction for multiple comparisons; Materials and methods). Second, we minimized movement effects by limiting the analysis window to a period that preceded 97% of lick onsets (0–150 ms from stimulus onset). We also included censor and grace periods (Materials and methods) to reduce the impact of compulsive licking, and excluded trials with licking during the

1 s window preceding stimuli. We note that motor-related signals need not always occur together with overt movement. For instance, subthreshold stimulation of the frontal eye fields in primates can drive V4 activity and mimic the effects of attention without causing eye movements (*Moore and Armstrong, 2003*; *Moore and Fallah, 2001*).

Responses to tactile stimuli within a respond-to-light block were significantly reduced but still observable in ALM (*Figure 1g* and *Figure 1—figure supplement 1g*). This suggests that gating of tactile information likely occurred in part prior to ALM (*Aruljothi et al., 2020*; *Borden et al., 2022*; *Zhang and Zagha, 2022*). In contrast, we did not observe visually evoked activity in ALM (*Figure 1g* and *Figure 1—figure supplement 1g and h*). This modality bias is consistent with the long-range connectivity between sensory and frontal areas. Specifically, the somatosensory cortex is connected to the motor cortex, whereas the visual cortex is connected to the anterior cingulate cortex (ACC) (*Zhang et al., 2016*). Also consistent with this anatomy is that we observed visually evoked activity in the anterior part of the ACC (*Figure 1—figure supplement 1i–k*; the AMM), and ACC has been shown to modulate V1 activity in rodents (*Zhang et al., 2014*).

In our task, the right or left water port was the rewarded port in a respond-to-touch block or a respond-to-light block, respectively. Although the mice could not anticipate stimulus types and licking responses during the ITI, there might be a subtle bias of posture and movement across blocks given the different positions of the rewarded port. To reduce the effects of movement bias on pre-stimulus activity (−100 to 0 ms), we removed trials with licking during a 1 s window before stimulus onset. Moreover, in a separate study using the same task (*Finkel et al., 2024*), high-speed video analysis demonstrated no significant differences in whisker motion between respond-to-touch and respond-to-light blocks in most (12 of 14) behavioral sessions.

We found that the neural subspaces containing population activity patterns were different during respond-to-touch and respond-to-light rules. Specifically, S1, S2, MM, and ALM showed lower subspace overlaps when calculated between tHit and tCR than when calculated between tHit and control (held-out) tHit (*Figure 4e*). These subspace differences could result from: (1) involvement of different sets of neurons; (2) differently signed changes in firing patterns (such as some neurons fire more and others fire less); and/or (3) differently scaled changes in firing patterns (such as the firing rates of some neurons do not change and the firing rates of other neurons increased twofold).

We found that how well neural subspaces were separated during tactile processing was associated with how well they were separated prior to the stimulus (*Figure 4f*). This result is consistent with a dynamical systems view of neural population processing, where initial conditions are of critical importance (*Vyas et al., 2020*; *Shenoy et al., 2013*). Bringing the population activity to an optimal initial state could allow the evolution of the population activity to produce the desired movement. Primate studies have shown that motor cortical areas implement this strategy to control movement (*Afshar et al., 2011*; *Churchland et al., 2010*; *Churchland et al., 2006b*). Here, we identified a potentially similar role for motor cortex activity states in the transformation of incoming sensory signals into actions in a rule-dependent manner. In addition, we found that not only motor but also sensory cortical areas had initial states that varied with the current rule. Indeed, pre-stimulus activity has been shown to encode rule information in primate visual cortex (*Jonikaitis et al., 2020*) and rodent auditory cortex (*Rodgers and DeWeese, 2014*). Overall, these findings indicate that setting up appropriate initial states could be a general strategy by which cortical networks integrate external inputs to achieve context-specific processing.

No-lick trials included misses, which could be caused by mice not being engaged in the task. While the majority of no-lick trials were correct rejections (respond-to-touch: 75%; respond-to-light: 76%), we treated no-licks as one of the available choices in our task and included them to calculate choice CDs (*Figure 4—figure supplement 1c, d, and f*). To ensure stable and balanced task engagement across task rules, we removed the last 20 trials of each session and used stimulus parameters that achieved similar behavioral performance for both task rules (*Figure 1d*; ~75% correct for both rules). However, we also calculated choice CDs using only right- and left-lick trials. In S1, S2, MM, and ALM, the choice CDs calculated this way were also not aligned well across task rules (*Figure 4—figure supplement 1h*), consistent with the results calculated from lick and no-lick trials (*Figure 4—figure supplement 1f*). Data were limited for this analysis, however, because mice rarely licked to the unrewarded water port (# of licks$_{unrewarded\ port}$ /# of licks$_{total}$, respond-to-touch: 0.13, respond-to-light: 0.11). These trials usually came from rule transitions (*Figure 5a*) and, in

some cases, were potentially caused by exploratory behaviors. These factors could affect choice CDs.

Inhibition of the motor cortical areas prior to stimulus delivery slightly but significantly impaired tactile detection in the respond-to-touch rule and visual detection in the respond-to-light rule (*Figure 6c* and *Figure 6—figure supplement 2g*). However, the ability to detect sensory stimuli was not completely abolished. This suggests that circuits for encoding the task rules may be redundant and/or other gating mechanisms may be involved (*Bari et al., 2019*). Indeed, the pre-stimulus activity in either MM and ALM could be used to decode the task rules, although we inhibited only one area at a time. Additionally, it has been shown that loops between ALM and subcortical regions including thalamus and cerebellum maintain persistent activity during short-term memory (*Gao et al., 2018*; *Guo et al., 2017*). It is possible that recurrent circuits across multiple brain areas contribute to holding rule information during ITI.

To test for a rule-specific function of pre-stimulus states, we used a simple tactile detection task to assess the potential effects of inhibition on sensory processing and lick production (*Figure 6e–g*). We found that inhibition of the pre-stimulus states of MM and ALM only reduced the detection sensitivity for tactile stimuli in the cross-modal selection task but not in the simple tactile detection task (*Figure 6c and f*), suggesting a role specific to flexible rule-dependent sensorimotor transformations. We balanced the behavioral performances in these tasks (~75% correct) via the adjustment of stimulus intensity to make the task difficulties similar. However, the effects of silencing cortex can also depend on factors that we did not probe, such as the time course of an area's task involvement (*Oude Lohuis et al., 2022*). To more precisely dissect the effects of perturbations on different cognitive processes in rule-dependent sensory detection, more complex behavioral tasks and richer behavioral measurements are needed in the future.

Pre-stimulus activity in MM and ALM showed a strong dependence on the current rule (*Figures 2, 3, and 5*), correlated with aspects of subsequent tactile processing (*Figure 4*), and was required for tactile detection during the cross-modal selection task (*Figure 6c*). These motor cortical areas are therefore likely to play an important role in the rule-dependent sensorimotor transformations occurring within cortical networks (*Siniscalchi et al., 2016*). A greater rule dependence of activity in motor compared with sensory areas is consistent with primate visual and somatosensory studies showing that attention effects become more prominent in higher-order areas (*Buffalo et al., 2010*; *Gregoriou et al., 2009*; *Hsiao et al., 1993*). Moreover, a number of studies in primates and rodents have shown that sensory-related responses in sensory cortical areas are modulated by motor cortical areas (*Petersen, 2019*; *Moore and Zirnsak, 2017*; *Moore and Armstrong, 2003*; *Lee et al., 2008*; *Zagha et al., 2013*). It is possible that MM and ALM received rule information from other brain regions such as the medial prefrontal cortex (*Bari et al., 2019*; *Guo et al., 2017*; *Anastasiades and Carter, 2021*; *Schmitt et al., 2017*) and send this information to S1 and S2 in the cross-modal selection task. Future work is needed to identify and characterize the neural circuits responsible for implementation, encoding, and updating of rules (*Mansouri et al., 2020*).

# Materials and methods

## Key resources table

| Reagent type (species) or resource | Designation | Source or reference | Identifiers | Additional information |
|---|---|---|---|---|
| Strain, strain background (*Mus musculus,* male and female) | *Pvalb*[Cre]: B6.129P2-Pvalb[tm1(cre)Arbr]/J | The Jackson Laboratory | 008069 | Materials and methods >Mice |
| Strain, strain background (*M. musculus,* male and female) | *Emx1*[Cre]: B6.129S2-Emx1[tm1(cre)Krj]/J | The Jackson Laboratory | 005628 | Materials and methods >Mice |
| Strain, strain background (*M. musculus,* male and female) | Ai32: B6.129S-Gt(ROSA)26Sor[tm32(CAG-COP4*H134R/EYFP)Hze]/J | The Jackson Laboratory | 012569 | Materials and methods >Mice |
| Strain, strain background (*M. musculus,* male and female) | *Slc32a1*[ChR2-EYFP]: B6.Cg-Tg (Slc32a1-COP4*H134R/ EYFP)8Gfng/J | The Jackson Laboratory | 014548 | Materials and methods >Mice |

*Continued on next page*

*Continued*

| Reagent type (species) or resource | Designation | Source or reference | Identifiers | Additional information |
|---|---|---|---|---|
| Software, algorithm | MATLAB | MathWorks | 2019a | https://www.mathworks.com/help/matlab/release-notes-R2019a.html |
| Software, algorithm | BControl software | Brody, Princeton University | | https://brodywiki.princeton.edu/bcontrol/index.php?title=Main_Page |
| Software, algorithm | Kilosort | *Pachitariu et al., 2016*; *Pachitariu et al., 2024* | Kilosort 1 | https://github.com/MouseLand/Kilosort |
| Other | Isoflurane | Penn Veterinary | VED1360CS | Materials and methods >Surgery |
| Other | Dental acrylic | Jet Repair Acrylic | L25-0335 | Materials and methods >Surgery |
| Other | Clear adhesive luting cement (C&B Metabond) | Parkell | S399 | Materials and methods >Surgery |
| Other | Silicone elastomer | World Precision Instruments | KWIK-CAST | Materials and methods >Surgery |
| Other | DiI Stain | Thermo Fisher Scientific | D282 | Materials and methods >Ephy. |
| Other | Piezo actuator | Piezo Systems | D220-A4-203YB | Materials and methods >Behav. tasks |
| Other | Piezo controller | Thorlabs | MDTC93B | Materials and methods >Behav. tasks |
| Other | LED | Thorlabs | M470F1 | Materials and methods >Behav. tasks |
| Other | Optic fiber | Thorlabs | M43L01, TM200FL1B | Materials and methods >Behav. tasks and Opto. inhibition |
| Other | Density filter | Thorlabs | NE530B | Materials and methods >Behav. tasks |
| Other | Stereotaxic apparatus (Mouse Gas Anesthesia Head Holder) | David Kopf Instruments | Model 923-B | Materials and methods >Surgery |
| Other | Silicon probe | Cambridge NeuroTech | ASSY-77 H3 | Materials and methods >Ephy. |
| Other | Intan recording system | Intan Technologies | RHD2000 | Materials and methods >Ephy. |
| Other | Laser, 473 nm | UltraLasers | DHOM-L-473–200 mW | Materials and methods >Opto. inhibition |
| Other | Acousto-optic modulator | QuantaTech | MTS110-A3-VIS | Materials and methods >Opto. inhibition |

## Mice

All procedures were performed in accordance with protocols approved by the Johns Hopkins University Animal Care and Use Committee (M018M187 and M021M195). Twelve mice (8 male, 4 female) were obtained by crossing *Pvalb$^{cre}$* lines (*Hippenmeyer et al., 2005*) (Jackson Labs: 008069) with Ai32 lines (*Madisen et al., 2012*) (Jackson Labs: 012569). Seven *Pvalb$^{cre}$*; Ai32 mice (5 male, 2 female) were trained to perform the cross-modal selection task and included in behavioral and optogenetic inhibition experiments. Five *Pvalb$^{cre}$*; Ai32 mice (3 male, 2 female) were trained to perform the tactile detection task and included in optogenetic inhibition experiments. Four male mice included in behavioral experiments were obtained by crossing *Emx1$^{cre}$* mice (*Gorski et al., 2002*) (Jackson Labs: 005628) with Ai32 mice. Two male mice included in behavioral experiments were heterozygous *Slc32a1$^{ChR2-EYFP}$* (Jackson Labs: 014548) (*Zhao et al., 2011*). Mice ranged in age from 2 to 5 months at the start of training. Mice were housed in a vivarium with a reverse light-dark cycle (12 hr each phase), and were singly housed after surgery and during behavioral experiments. Details of assignment to different experimental conditions are listed in *Table 1*.

**Table 1.** Experimental subjects.
Tabulated metadata for each mouse, including appearances in each figure.

| Animal ID | Genotype | Sex | Date of Birth | Test Session Dates | Figure Appearances |
|---|---|---|---|---|---|
| EF0147 | Emx1cre; Ai32 | male | 180627 | 190204–190,206 | *Figure 1d, e, g and h; Figure 2b–f, Figure 3b–e; Figure 4; Figure 5; Figure 1—figure supplement 1h, k; Figure 1—figure supplement 2; Figure 3—figure supplement 1; Figure 4—figure supplement 1; Figure 5—figure supplement 1* |
| EF0148 | Emx1cre; Ai32 | male | 180627 | 190228–190,305 | *Figure 1d, e, g and h; Figure 2b–f, Figure 3b–e; Figure 4; Figure 5; Figure 1—figure supplement 1c, i, k; Figure 1—figure supplement 2; Figure 3—figure supplement 1; Figure 4—figure supplement 1; Figure 5—figure supplement 1* |
| EF0150 | Emx1cre; Ai32 | male | 180627 | 190223–190,318 | *Figure 1d, e, g and h; Figure 2b–f, Figure 3b–e; Figure 4; Figure 5; Figure 1—figure supplement 1j, k; Figure 1—figure supplement 2; Figure 3—figure supplement 1; Figure 4—figure supplement 1; Figure 5—figure supplement 1* |
| EF0151 | Emx1cre; Ai32 | male | 180627 | 190506 | *Figure 1; Figure 2b–f, Figure 3b–e; Figure 4; Figure 5; Figure 1—figure supplement 1g, k; Figure 1—figure supplement 2; Figure 3—figure supplement 1; Figure 4—figure supplement 1; Figure 5—figure supplement 1* |
| YT053 | Pvalbcre; Ai32 | male | 180211 | 181115–181,207 | *Figure 6c, d, i; Figure 6—figure supplement 1b, c, e, f; Figure 6—figure supplement 2* |
| JL005 | Slc32a1ChR2-EYFP | male | 181121 | 190716–190,723 | *Figure 1d, e, g and h; Figure 2b–f, Figure 3b–e; Figure 4; Figure 5; Figure 1—figure supplement 1e, k; Figure 1—figure supplement 2; Figure 3—figure supplement 1; Figure 4—figure supplement 1; Figure 5—figure supplement 1* |
| YT071 | Slc32a1ChR2-EYFP | male | 180726 | 190721–190,722 | *Figure 1d, e, g and h; Figure 2; Figure 3b–e; Figure 4; Figure 5; Figure 1—figure supplement 1f, k; Figure 1—figure supplement 2; Figure 3—figure supplement 1; Figure 4—figure supplement 1; Figure 5—figure supplement 1* |
| YT080 | Pvalbcre; Ai32 | male | 181112 | 190503–190514; 190827–190,828 | *Figure 1d, e, g and h; Figure 2b–f, Figure 3b–e; Figure 4; Figure 5; Figure 6c, d, i; Figure 1—figure supplement 1k; Figure 1—figure supplement 2; Figure 3—figure supplement 1; Figure 4—figure supplement 1; Figure 5—figure supplement 1; Figure 6—figure supplement 1b, c, e, f; Figure 6—figure supplement 2* |
| YT081 | Pvalbcre; Ai32 | male | 181112 | 190418–190505; 190826–190,908 | *Figure 1d, e, g and h; Figure 2b–f, Figure 3b–e; Figure 4; Figure 5; Figure 6c, d and i; Figure 1—figure supplement 1b, k; Figure 1—figure supplement 2; Figure 3—figure supplement 1; Figure 4—figure supplement 1; Figure 5—figure supplement 1; Figure 6—figure supplement 1b, c, e, f; Figure 6—figure supplement 2* |
| YT083 | Pvalbcre; Ai32 | female | 190401 | 191205–200108; 200123 | *Figure 1d, e, g and h; Figure 2b–f, Figure 3b–e; Figure 4; Figure 5; Figure 6c, d and i; Figure 1—figure supplement 1k; Figure 1—figure supplement 2; Figure 3—figure supplement 1; Figure 4—figure supplement 1; Figure 5—figure supplement 1; Figure 6—figure supplement 1b, c, e, f; Figure 6—figure supplement 2* |
| YT084 | Pvalbcre; Ai32 | female | 190620 | 191020–200103; 200124–200,210 | *Figure 1d, e, g and h; Figure 2b–f, Figure 3b–e; Figure 4; Figure 5; Figure 6c, d and i; Figure 1—figure supplement 1a, k; Figure 1—figure supplement 2; Figure 3—figure supplement 1; Figure 4—figure supplement 1; Figure 5—figure supplement 1; Figure 6—figure supplement 1b, c, e, f; Figure 6—figure supplement 2* |
| YT085 | Pvalbcre; Ai32 | male | 190620 | 191205–200108; 200308–200,320 | *Figure 1d, e, g and h; Figure 2b–f, Figure 3b–e; Figure 4; Figure 5; Figure 6c, d and i; Figure 1—figure supplement 1d, k; Figure 1—figure supplement 2; Figure 3—figure supplement 1; Figure 4—figure supplement 1; Figure 5—figure supplement 1; Figure 6—figure supplement 1b, c, e, f; Figure 6—figure supplement 2* |
| YT086 | Pvalbcre; Ai32 | male | 190620 | 191205–200110; 200314–200,320 | *Figure 1d, e, g and h; Figure 2b–f, Figure 3b–e; Figure 4; Figure 5; Figure 6c, d and i; Figure 1—figure supplement 1b, k; Figure 1—figure supplement 2; Figure 3—figure supplement 1; Figure 4—figure supplement 1; Figure 5—figure supplement 1; Figure 6—figure supplement 1b, c, e, f; Figure 6—figure supplement 2* |
| YT091 | Pvalbcre; Ai32 | male | 200830 | 201124–201,211 | *Figure 6f, g, j; Figure 6—figure supplement 1h, i, k, l* |
| YT092 | Pvalbcre; Ai32 | female | 200830 | 201126–201,214 | *Figure 6f, g, j; Figure 6—figure supplement 1h, i, k, l* |
| YT093 | Pvalbcre; Ai32 | male | 200919 | 210102–210,106 | *Figure 6f, g, j; Figure 6—figure supplement 1h, i, k, l* |
| YT094 | Pvalbcre; Ai32 | male | 200919 | 201228–210,103 | *Figure 6f, g, j; Figure 6—figure supplement 1h, i, k, l* |

*Table 1 continued on next page*

*Table 1 continued*

| Animal ID | Genotype | Sex | Date of Birth | Test Session Dates | Figure Appearances |
|---|---|---|---|---|---|
| YT095 | Pvalb^cre; Ai32 | female | 200919 | 201230–210,116 | *Figure 6f, g, j; Figure 6—figure supplement 1h, i, k, l* |

## Behavioral tasks

All behavioral experiments were conducted with head-fixed mice during the dark phase. Behavioral apparatus was controlled by BControl software (C. Brody, Princeton University). Four to 7 days after a headpost implantation and 7–14 days before behavioral training, mice were allowed 1 ml of water daily until reaching ~80% of their starting body weight. On training days, mice were allowed to perform until sated (~1 hr/day) and were weighed before and after each session to determine the amount of water consumed. Additional water was given if mice consumed <0.3 ml of water in order to maintain a stable body weight. On days when their behavior was not tested, they received 1 ml of water.

## Cross-modal sensory selection task

The cross-modal sensory selection training consists of two stages. Mice were first trained to perform tactile and visual detection separately, then trained on the cross-modal selection task where tactile and visual stimuli were randomly interleaved.

In the first 1–2 sessions, mice were acclimated to head fixation in the behavioral apparatus while being given free access to water via two reward ports located 6–10 mm and ~35 degrees to the left and right of the mouse midline. In subsequent sessions, mice were randomly assigned to start with tactile or visual detection training. After the hit rate of one modality reached >70% (~3 days; hit rate = 100*(# hits)/(# hits + # misses)), stimulus detection training of the other modality began. For tactile detection training, a single whisker (always on the right whisker pad) was threaded into a glass pipette attached to a piezo actuator (D220-A4-203YB, Piezo Systems), which was driven by a piezo controller (MDTC93B, Thorlabs). Approximately 1.5 mm of whisker remained exposed at the base. All whiskers except the target whisker were trimmed to near the base. Mice were given a drop of water (~6 µl) for licking to the right reward port in response to a tactile stimulus (1 s sinusoidal deflections at 40 Hz, ~1400 degrees/s) during an answer period (0.1–2 s from stimulus onset). For visual detection training, mice were rewarded for licking to the left water port in response to a visual stimulus. Each visual stimulus comprised 470 nm light (1 s flash at ~5 mW) generated by an LED (M470F1 LED driven by LEDD1B, Thorlabs) and emitted from the tip of an optic fiber (105 µm diameter, 0.22 NA; M43L01, Thorlabs) positioned 5.5 cm away from the tip of the mouse's nose along its midline. The intensity of visual stimulus can be adjusted by a density filter (NE530B, Thorlabs). To reduce compulsive licking, licks that occurred within a 'grace period' (0–0.1 s from stimulus onset) were not rewarded. Licks occurring in a 'censor period' (−0.2–0 s from stimulus onset) resulted in the withholding of the stimulus presentation for that trial and no reward or punishment. Trials with licks occurring in the grace and censor periods were omitted from analysis. In all sessions, ambient white noise (cutoff at 40 kHz, ~80 dB SPL) was played to mask any potential sound associated with movement of the piezo stimulator.

After the stimulus detection training, mice were trained to perform the cross-modal selection task. Tactile and visual stimuli were randomly interleaved (subject to a limit of 4 consecutive trials of the same type) and trials were separated by a random interval (3.5 s fixed interval+random interval drawn from an exponential distribution with mean 4 s). Trials were grouped into either respond-to-touch or respond-to-light blocks (54–66 trials per block from a uniform distribution with mean 60 trials). Each session randomly began with one of two block types, and the block types subsequently alternated multiple times (4–6 blocks per session). In respond-to-touch blocks, mice were rewarded with a drop of water if they licked the right reward port following tactile but not visual stimuli. In respond-to-light blocks, mice were rewarded with a drop of water for licking the left reward port following visual but not tactile stimuli. The answer, grace, and censor periods were as described above for the stimulus detection training.

Four trial outcomes were defined based on block types, sensory stimuli, and responses (*Figure 1c*). Trials in which mice licked to the correct reward port following tactile stimuli in respond-to-touch blocks or visual stimuli in respond-to-light blocks were scored as 'hit' trials. Failures to lick to the correct port

after tactile stimuli in respond-to-touch blocks or visual stimuli in respond-to-light blocks were scored as 'miss' trials. Licks to either reward port after tactile stimuli in respond-to-light blocks, visual stimuli in respond-to-touch blocks, or to the incorrect reward port after either stimulus type, resulted in 'false alarm' trials. Trials in which mice correctly withheld licks after tactile stimuli in respond-to-light blocks, or after visual stimuli in respond-to-touch blocks, were scored as 'correct rejection' trials. Performance was quantified as percent correct: 100*(# hits + # correct rejections)/(# of trials total).

In an initial stage of cross-modal selection training (~7 sessions), a drop of water from the rewarded port was automatically released following 80% behaviorally relevant stimuli. Subsequently, automatically released water only occurred on the 9th trial after a block switch. Once performance reached >70% of trials correct, task difficulty was gradually increased by reducing stimulus intensity and duration. In a final stage of training, faint stimuli (tactile: 0.15 s sinusoidal deflections at 20 Hz, ~800 degrees/s; visual: 0.15 s flash at ~3 μW) were used to increase cognitive load and to result in error trials for analysis. Mice were considered trained when performance reached >70% correct for at least 3 consecutive days. After reaching this performance criterion, mice proceeded with test sessions. Seven *Pvalb^cre*; Ai32 mice performed the cross-modal task during inhibition experiments, and six of them continued for electrophysiology recordings. Other transgenic mice were given test sessions for electrophysiology recordings but not optogenetic experiments.

Behavioral sessions lasted until mice were sated. To ensure stable engagement, the last 20 trials of each session were removed from further analysis. In addition, sessions were omitted from analysis if overall performance was <60% correct, at least one of block performances (respond-to-touch or respond-to-light blocks) was <55% correct, or at least one of hit rates (tactile or visual hits) was <35%. Three sessions in total were removed for these reasons (from two mice).

## Tactile detection task

Head-fixed mice were trained to perform a Go/NoGo tactile detection task. On Go trials, the whisker was deflected (0.15 s sinusoidal deflections at 20 Hz, ~600 degrees/s). If mice licked the right reward port following a tactile stimulus, a drop of water was released and it was scored as a hit trial. If mice failed to respond to a tactile stimulus, it was scored as a miss trial. On NoGo trials, the target whisker was not deflected. If mice licked during the answer period, it was scored as a false alarm. If mice withheld licking, it was scored as a correct rejection. Go and NoGo trials were randomly interleaved (subject to a limit of 4 consecutive trials of the same type), and no trial-start cue was presented. The answer, grace, and censor periods were as described above for the cross-modal selection task. Tactile stimuli of the tactile detection task were slightly weaker compared with the cross-modal selection task in order to control task difficulties by making behavioral performance similar (~75% correct).

Similar to the cross-modal selection task, the last 20 trials in each session were excluded, and sessions with performance <60% correct or tHit rate <35% were removed from subsequent analysis. Five *Pvalb^cre*; Ai32 mice performed the tactile detection task during inhibition experiments.

## Surgery

Prior to behavioral testing, mice were implanted with clear-skull caps (*Guo et al., 2014*) and metal headposts designed to expose a large area of the dorsal surface of the skull. During surgery, mice were anesthetized under isoflurane (1–2% in $O_2$; Surgivet) and mounted in a stereotaxic apparatus (David Kopf Instruments) with a thermal blanket (Harvard Apparatus). Mice were given a subcutaneous injection of Marcaine or Lidocaine for local analgesia and an intraperitoneal injection of Ketoprofen to reduce inflammation. The scalp and periosteum over the dorsal surface of the skull were removed. To expose S2 on the left hemisphere, the left temporal muscle was detached and the bone ridge at the temporal-parietal junction was thinned using a dental drill. Headposts were fixed to the skull over the lambda structure using clear adhesive luting cement (C&B Metabond Quick Adhesive Cement System; Parkell). A thin layer of clear cement followed by an additional layer of cyanoacrylate glue (Krazy Glue) was applied to the entire surface of the exposed skull, leaving it largely transparent. To protect the clear skull from scratching, a silicone elastomer (Kwik-Cats) was applied prior to optogenetic experiments.

Intrinsic signal imaging (ISI) was used to guide the whisker parts of S1 and S2 for neural recordings and optogenetic experiments (*Masino et al., 1993*; *O'Connor et al., 2013*). Mice were lightly

anesthetized with isoflurane (0.5–1%) and chlorprothixene. The C2 or C3 whisker was stimulated with a Piezo at 10 Hz. Since S2 is close to the auditory cortex, white noise was played during imaging.

For silicon probe recording, a small craniotomy (~1 mm in diameter) over the recording site (always on the left hemisphere) was made (S1 and S2 determined by ISI; MM: 1.5 mm anterior, 1.0 mm lateral; ALM: 2.5 mm anterior, 1.5 mm lateral to bregma). The dental acrylic and skull was thinned using a dental drill and the remaining bone was removed with a tungsten needle or forceps. A separate, smaller craniotomy (~0.6 mm in diameter) on the right hemisphere was made for implantation of a ground screw (0.6 mm anterior, 3.0 mm lateral to bregma; S1 trunk region). Additional craniotomies were usually made in new locations after finishing recordings in previous ones (12 mice; 1–4 recording sites per mouse).

## Electrophysiology and data preprocessing

Linear 64-channel probes (H3, Cambridge NeuroTech) were coated with DiI (saturated) or DiD (5–10 mg/ml) to histologically verify the site of recording post hoc. The silicon probe was inserted into the cortex either vertically (for MM and ALM) or at ~40 degrees from vertical (for S1 and S2). After probe insertion, the brain was covered with a layer of 1.5% agarose and ACSF and was left for ~10 min prior to recording.

Neural signals and behavioral timestamps were recorded using an Intan system (RHD2000 series multi-channel amplifier chip; Intan Technologies). Neural signals were sampled at 30 kHz. Kilosort was used for spike sorting (*Pachitariu et al., 2016*) and spike clusters were manually curated using Phy. Units were excluded from further analysis if the rate of inter-spike-interval violations within a 1.5ms window was >0.5%, L-ratios were >0.1, the presence of spikes was <90% of the whole session, the cumulative drift of spike depth was >40 μm (Unit Quality Metrics, Allen Institute).

For analyses about stimulus-evoked responses, neural spike rates were calculated in 10 ms bins and smoothed with a Gaussian kernel (50 ms). For analyses about pre-stimulus activity, neural spike rates were calculated in 100 ms bins without smoothing. Spike rates of simultaneously recorded neurons were normalized for all population-level analyses including LDA and PCA. We used soft normalization to make activity in a roughly unity range and to reduce the impact of units with low firing rate (normalized response = (response–mean(response))/(range(response)+5)) (*Churchland et al., 2012*; *Russo et al., 2020*). In addition, to minimize movement effects on neural activity during the pre-stimulus window (−1 to 0 s from stimulus onset), trials with licking occurring in this window were removed (~25%).

## Optogenetic inhibition

*Pvalb^cre*; Ai32 mice implanted with clear-skull caps were given optogenetic inhibition experiments after behavioral task training (cross-modal selection or tactile detection). Laser stimuli (473 nm; MBL-III-473-100, Ultralasers) were directed to the brain via optic fibers (200 μm diameter, 0.22 NA; TM200FL1B, Thorlabs) positioned over (~2 mm above) the cortical areas bilaterally (8–10 mW each side). The intensity of laser stimuli was controlled by an acousto-optic modulator (MTS110-A3-VIS, QuantaTech). For S1 and S2, the left whisker areas were guided by ISI (as described above) and the right whisker areas were determined as the symmetric positions. Other targeted areas on the dorsal cortex included MM (1.5 mm anterior, 1.0 mm lateral to bregma), ALM (2.5 mm anterior, 1.5 mm lateral), AMM (2.5 mm anterior, 0.5 mm lateral), and PPC (1.94 mm posterior, 1.6 mm lateral). Sham sessions were identical to optogenetic inhibition sessions except that the dorsal cortex was covered by blackout cloth in order to not inhibit any brain areas. For each session, one of the cortical areas or the sham condition was randomly assigned for inhibition. A cone, blackout cloth and tape were used to shield the mouse's eyes from scattered light due to the laser.

For the cross-modal sensory selection task, laser stimuli were delivered to the targeted brain areas in ~30% of tactile and visual stimulus trials. In around half of these trials, optogenetic inhibition occurred before stimulus onset to suppress baseline activity (−0.8 to 0 s from stimulus onset, 40 Hz sinusoidal waveform with a 0.1 s linearly modulated ramp-down at the end). In the other half of these trials, optogenetic inhibition began simultaneously with the stimulus onset to suppress sensory-evoked activity (0–2 s from stimulus onset, 40 Hz sinusoidal waveform with a 0.2 s linearly modulated ramp-down at the end). Additionally, in a subset of trials (~20%), laser stimuli were delivered alone.

These 'laser-only' trials consisted of short (0.8 s) and long (2 s) trains of laser pulses that are identical to the laser stimuli in pre-stimulus-onset and post-stimulus-onset conditions respectively.

For the tactile detection task, laser stimuli were delivered in ~30% of trials. Among these laser trials, Go trials consist of approximately half pre-stimulus-onset and half post-stimulus-onset inhibition. Since there was no tactile stimulus in NoGo trials, short (0.8 s) and long (2 s) laser stimuli were delivered before and after trial onset respectively. Laser stimuli are identical to those used in the cross-modal selection task.

Baseline behavioral performance was measured by trials without laser stimuli and used to determine if a session passed the criteria of good performance (as described in the Behavioral tasks section). In addition, sessions with laser catch rates >75% or >hit rates were removed from analysis because a high laser catch rate indicates that mice detected laser stimuli instead of tactile or visual stimuli (laser catch rate = 100*(# of laser-only trials in which licking occurred)/(# of laser-only trials total)).

### Single-neuron discrimination analyses
### ROC analysis
ROC analysis was used to calculate how well trial-by-trial activity of a single neuron could discriminate certain conditions (e.g. tHit vs tCR). The AUC represents the performance of an ideal observer in discriminating trials based on these conditions (MATLAB 'perfcurve'). For discriminability of touch-evoked activity between task rules (*Figure 1h*), tactile correct trials were split into tHit (respond-to-touch) and tCR (respond-to-light). The analysis window was the first 150 ms after stimulus onset to minimize any movement effects resulting from licking. A Bonferroni corrected 95% confidence interval for AUC was obtained by bootstrap. For each time bin (10 ms), if its 95% CI did not include the chance level (0.5), it was considered significant. We defined a unit as showing significant tHit-tCR selectivity when three consecutive time bins (>30 ms) of AUC values were significant.

For discriminability of pre-stimulus activity between task rules (*Figure 2c*), correct trials were split based on block types (respond-to-touch: tHit and visual correct rejections; respond-to-light: visual hits and tCR). The analysis window was the 100 ms window before stimulus onset. For discriminability of pre-stimulus activity between stimulus types (*Figure 2d*), correct trials were split based on stimulus types rather than block types (tactile: tHit and tCR; visual: visual hits and visual correct rejections). For discriminability of sensory-evoked activity between stimulus types (*Figure 2e*), correct trials were split based on stimulus types, and the analysis window was the first 100 ms after stimulus onset rather than before stimulus onset. For *Figure 2c–e*, the analysis window was one time bin (100 ms). If the Bonferroni corrected 95% CI for this time bin did not include the chance level (0.5), it was considered significant.

### PSTH-based permutation test
To determine whether water reward affected touch-evoked activity in the cross-modal selection task, we compared the mean PSTHs for tHit and for tactile false alarms in which mice licked to the right water port following a tactile stimulus in the respond-to-light blocks (*Figure 1c*). For each neuron, the Euclidean distance between the mean PSTHs for tHit and tactile false alarms was calculated (0–250 ms from stimulus onset). We then performed a permutation test on whether this Euclidean distance was significantly different from zero (*Foffani and Moxon, 2004*; *O'Connor et al., 2010*). A p-value was then calculated using the distribution of resampled Euclidean distances. Significance was determined at the alpha = 0.05 level after Bonferroni correction for the number of neurons.

### Population decoding analyses
We used LDA (MATLAB 'fitcdiscr') to measure how well population activity from simultaneously recorded neurons could decode (1) task rules (respond-to-touch vs respond-to-light) prior to stimulus delivery (−100 to 0 ms from the stimulus onset), (2) stimulus types (tactile vs visual stimuli) prior to stimulus delivery, and (3) stimulus types after stimulus onset (0–100 ms). All correct trials were used and classification accuracy was obtained using 10-fold cross-validation (*Figure 3* and *Figure 3—figure supplement 1c and d*). In addition, SVM (*Figure 3—figure supplement 1a*; MATLAB 'fitcsvm') and Random Forests (*Figure 3—figure supplement 1b*; MATLAB 'TreeBagger' with 500 trees) were used to discriminate task rules prior to stimulus delivery. The shuffled data was generated by shuffling the labels for individual trials (e.g. block types).

We also applied LDA to determine how the pre-stimulus states shifted during rule transitions (*Figure 5* and *Figure 5—figure supplement 1*). We used 90% of the correct trials as training data for task rules and the held-out 10% of correct trials to classify trials as having come from respond-to-touch or respond-to-light blocks. We also separately classified trials occurring in the 'early transition' and 'late transition' periods as having come from one or the other of the block types.

## Distance between neural trajectories

We calculated the distance between tHit and tCR trajectories to determine how these trajectories diverged (*Figure 4a and b*). For each session, we performed a PCA using the trial-averaged tHit and tCR population spike rate responses (−100 to 150 ms from stimulus onset). Population responses for individual tHit and tCR trials were projected onto the top three PC space. For each pair of tHit and tCR trials, the Euclidean distances between the neural states of tHit and tCR trajectories at each time point were calculated. The distances between tHit and tCR trajectories were averaged across these pairs in each session.

To investigate the relationship between a difference in pre-stimulus activity and a difference in subsequent sensory-evoked activity, the distances between tHit and tCR trajectories from all recording sessions (total 40 sessions; S1 [10], S2 [8], MM [9], ALM [13]) were ranked based on the distances before stimulus delivery (−100 to 0 ms). The mean tHit-tCR distances after stimulus onset (0–150 ms) between the top and bottom 50% groups were compared using a permutation test (*Figure 4c*). Specifically, we calculated the Euclidean distance between the mean tHit-tCR distances for these two groups. The group labels were then randomly shuffled, and new mean tHit-tCR distances of the shuffled groups were obtained. The Euclidean distance between these shuffled mean tHit-tCR distances was calculated. This shuffling procedure was repeated 1000 times, and then the p-value was calculated (one-tailed; null hypothesis: no difference; distance ≥ 0).

## Subspace overlap

The subspace overlap between tHit and tCR trials was obtained through their variance alignment (*Figure 4d–f*). For each session, the trial-averaged tHit activity was used to perform a PCA (0–150 ms from stimulus onset). The trial-averaged tCR activity was projected onto the top three PC space (tCR$_{tHit-subspace}$), and the variance explained was calculated. For normalization, a separated PCA was performed on the trial-averaged tCR activity, and its own (tCR$_{tCR-subspace}$) variance explained was calculated. The subspace overlap was defined as the ratio of the variance explained of tCR$_{tHit-subspace}$ to the variance explained of tCR$_{tCR-subspace}$. We chose the top three PCs because most of the variances of tHit$_{tHit-subspace}$ (~90%) and tCR$_{tCR-subspace}$ (~85%) were captured.

To test if the subspace for processing tactile signals significantly changed under different rules, we compared the subspace overlap between tHit and tCR trials with a control group. Specifically, we randomly assigned tHit trials into equal sized reference and control groups. The tCR and tHit control groups were projected to the PC space of the tHit reference group, and their subspace overlaps were compared (*Figure 4e*). To calculate the separation of subspaces prior to stimulus delivery, pre-stimulus activity in tCR trials (−100 to 0 ms from stimulus onset) was projected to the PC space of the tHit reference group and the subspace overlap was calculated. In this analysis, we used tHit activity during stimulus delivery (0–150 ms from stimulus onset) to obtain reliable PCs. In addition, the subspace overlap could be overestimated when there were only few neurons in a session (low dimensionality). To avoid this issue, sessions having less than 10 units were excluded from this analysis.

## Stimulus and choice CDs

For each session, *n* simultaneously recorded neurons created an *n* dimensional space. A CD is defined as an *nx1* vector that maximally separates the neural trajectories for different conditions (*Li et al., 2016*; *Yang et al., 2022*). For example, to estimate a stimulus CD in respond-to-touch blocks, we used trial-averaged trajectories for tactile ($x_{tactile} = (x_{tactile-right-lick} + x_{tactile-no-lick})/2$) and visual ($x_{visual} = (x_{visual-right-lick} + x_{visual-no-lick})/2$) trials and calculated their difference at each time point ($\nu_t = x_{tactile} - x_{visual}$). We then averaged $\nu_t$ during the analysis window (0–150 ms from stimulus onset) to obtain the stimulus CD.

In *Figure 4—figure supplement 1*, the trial types used to calculate stimulus and choice CDs were:

| Figure a–f | Respond-to-touch blocks | Respond-to-light blocks |
|---|---|---|
| Stimulus CD | Tactile (tactile-right-lick, tactile-no-lick) vs Visual (visual-right-lick, visual-no-lick) | Tactile (tactile-left-lick, tactile-no-lick) vs Visual (visual-left-lick, visual-no-lick) |
| Choice CD | Right-lick (tactile-right-lick, visual-right-lick) vs No-lick (tactile-no-lick, visual-no-lick) | Left-lick (tactile-left-lick, visual-left-lick) vs No-lick (tactile-no-lick, visual-no-lick) |

| Figure g and h | Respond-to-touch blocks | Respond-to-light blocks |
|---|---|---|
| Stimulus CD | Tactile (tactile-right-lick, tactile-left-lick) vs Visual (visual-right-lick, visual-left-lick) | Tactile (tactile-right-lick, tactile-left-lick) vs Visual (visual-right-lick, visual-left-lick) |
| Choice CD | Right-lick (tactile-right-lick, visual-right-lick) vs Left-lick (tactile-left-lick, visual-left-lick) | Right-lick (tactile-right-lick, visual-right-lick) vs Left-lick (tactile-left-lick, visual-left-lick) |

To test if stimulus (choice) CDs changed with the task rules, we calculated the dot product between the stimulus (choice) CD in respond-to-touch blocks and the stimulus (choice) CD in respond-to-light blocks. The CDs here are unit vectors, so the magnitude of the dot product ranges from 0 (orthogonal) to 1 (aligned).

## Stimulus sensitivity

For the cross-modal sensory selection task, the detection sensitivity for tactile stimuli was calculated as the difference of tHit rate and visual false alarm rate during the respond-to-touch blocks. The tHit rate was the probability of licking right in response to tactile stimuli, and the visual false alarm rate was the probability of licking right in response to visual stimuli. Correspondingly, the detection sensitivity for visual stimuli was determined by the difference of visual hit rate and tactile false alarm rate during the respond-to-light blocks. The visual hit rate was the probability of licking left in response to visual stimuli, and the tactile false alarm rate was the probability of licking left in response to tactile stimuli.

For the tactile detection task, the detection sensitivity for tactile stimuli was calculated as the difference of hit rate and false alarm rate. The hit rate was the probability of licking in the stimulus trials (Go trials), and the false alarm rate was the probability of licking in the no-stimulus trials (NoGo trials).

## Statistics

We report data as mean ± standard error of the mean (s.e.m.) except where noted. Statistical tests were two-tailed unless otherwise noted. We made the Bonferroni correction for multiple comparisons across neurons in each cortical area (*Figures 1h and 2c–e*).

We calculated confidence intervals using a non-parametric hierarchical bootstrap method (*Saravanan et al., 2019*) to simulate the data generation process and to incorporate variability at different levels including mice, sessions, neurons, and trial types (1000 iterations). For population decoding analysis (*Figures 3–5*), statistical tests were performed across sessions (e.g. a mean classification accuracy for task rules across sessions). For behavioral analysis during optogenetic inhibition (*Figure 6*), statistical tests were performed across mice (e.g. a mean tactile sensitivity across mice).

## Acknowledgements

We thank William Olson for comments on the manuscript; Emily Lubin and Jae Hun Lee for technical assistance; and Genki Minamisawa for technical advice. We thank Luiz Tauffer and Ben Dichter from CatalystNeuro for converting our data to the Neurodata Without Borders format. This work was supported by a Government Scholarship to Study Abroad from the Ministry of Education of Taiwan to Y-TC, Seed Grant S-2021-GR-045 from The Kavli Foundation to DHO, and NIH grants R01NS089652 and 1R01NS104834-01 to DHO.

## Additional information

### Funding

| Funder | Grant reference number | Author |
|---|---|---|
| National Institute of Neurological Disorders and Stroke | R01NS089652 | Daniel H O'Connor |
| National Institute of Neurological Disorders and Stroke | 1R01NS104834-01 | Daniel H O'Connor |
| Kavli Foundation | S-2021-GR-045 | Daniel H O'Connor |
| Ministry of Education | Government Scholarship to Study Abroad | Yi-Ting Chang |

The funders had no role in study design, data collection and interpretation, or the decision to submit the work for publication.

### Author contributions

Yi-Ting Chang, Conceptualization, Data curation, Formal analysis, Funding acquisition, Investigation, Methodology, Writing - original draft, Writing - review and editing; Eric A Finkel, Conceptualization, Methodology; Duo Xu, Software, Methodology; Daniel H O'Connor, Conceptualization, Resources, Supervision, Funding acquisition, Writing - original draft, Writing - review and editing

### Author ORCIDs

Yi-Ting Chang ⓘ http://orcid.org/0000-0001-5327-250X
Duo Xu ⓘ http://orcid.org/0000-0002-8259-8688
Daniel H O'Connor ⓘ http://orcid.org/0000-0002-9193-6714

### Ethics

All procedures were performed in accordance with protocols approved by the Johns Hopkins University Animal Care and Use Committee (M018M187 and M021M195).

Reviewer #1 (Public Review): https://doi.org/10.7554/eLife.92620.3.sa1
Reviewer #2 (Public Review): https://doi.org/10.7554/eLife.92620.3.sa2
Reviewer #3 (Public Review): https://doi.org/10.7554/eLife.92620.3.sa3
Author response https://doi.org/10.7554/eLife.92620.3.sa4

---

## Additional files

### Supplementary files

• MDAR checklist

### Data availability

Data in the Neurodata Without Borders (NWB) format is available at DANDI archive (https://doi.org/10.48324/dandi.000232/0.240510.2038). The processed data is available at https://doi.org/10.5281/zenodo.11176244. The MATLAB scripts used to analyze the data are available at https://github.com/YitingChang/cross_modal_task (copy archived at *Chang, 2024*).

The following datasets were generated:

| Author(s) | Year | Dataset title | Dataset URL | Database and Identifier |
|---|---|---|---|---|
| Chang Y-T, O'Connor DH | 2024 | Rule-based modulation of a sensorimotor transformation across cortical areas | https://doi.org/10.48324/dandi.000232/0.240510.2038 | DANDI, 10.48324/dandi.000232/0.240510.2038 |

*Continued on next page*

*Continued*

| Author(s) | Year | Dataset title | Dataset URL | Database and Identifier |
|---|---|---|---|---|
| Chang Y-T, O'Connor DH | 2024 | Datasets of Chang et al 2024 | https://doi.org/10.5281/zenodo.11176244 | Zenodo, 10.5281/zenodo.11176244 |

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
