## [Editor Report · eLife assessment]

This **important** work advances our understanding of how brains flexibly gate actions in different contexts, a topic of great interest to the broader field of systems neuroscience. Recording neural activity from several sensory and motor cortical areas along a sensorimotor pathway, the authors found that preparatory activity in motor cortical areas of the mouse depends on the context in which an action will be carried out, consistent with previous theoretical and experimental work. Furthermore, the authors provide causal evidence that these changes support flexible gating of actions. The carefully carried out experiments were analyzed using state-of-the-art methodology and provide **convincing** conclusions.

---

## [Referee Report · Reviewer #1 (Public Review)]

Summary:

Using a cross-modal sensory selection task in head-fixed mice, the authors attempted to characterize how different rules reconfigured representations of sensory stimuli and behavioral reports in sensory (S1, S2) and premotor cortical areas (medial motor cortex or MM, and ALM). They used silicon probe recordings during behavior, a combination of single-cell and population-level analyses of neural data, and optogenetic inhibition during the task.

Strengths:

A major strength of the manuscript was the clarity of the writing and motivation for experiments and analyses. The behavioral paradigm is somewhat simple but well-designed and well-controlled. The neural analyses were sophisticated, clearly presented, and generally supported the authors' interpretations. The statistics are clearly reported and easy to interpret. In general, my view is that the authors achieved their aims. They found that different rules affected preparatory activity in premotor areas, but not sensory areas, consistent with dynamical systems perspectives in the field that hold that initial conditions are important for determining trial-based dynamics.

I think this is a well-performed, well-written and interesting study that shows differences in rule representations in sensory and premotor areas, and finds that rules reconfigure preparatory activity in motor cortex to support flexible behavior.

---

## [Referee Report · Reviewer #2 (Public Review)]

Summary:

Chang et al. investigated neuronal activity firing patterns across various cortical regions in an interesting context-dependent tactile vs visual detection task, developed previously by the authors (Chevee et al., 2021; doi: 10.1016/j.neuron.2021.11.013). The authors report the important involvement of a medial frontal cortical region (MM, probably a similar location to wM2 as described in Esmaeili et al., 2021 & 2022; doi: 10.1016/j.neuron.2021.05.005; doi: 10.1371/journal.pbio.3001667) in mice for determining task rules.

Strengths:

The experiments appear to have been well carried out and the data well analysed. The manuscript clearly describes the motivation for the analyses and reaches clear and well-justified conclusions. I find the manuscript interesting and exciting!

Weaknesses:

I did not find any major weaknesses.

---

## [Referee Report · Reviewer #3 (Public Review)]

Summary:

This study examines context-dependent stimulus selection by recording neural activity from several sensory and motor cortical areas along a sensorimotor pathway, including S1, S2, MM, and ALM. Mice are trained to either withhold licking or perform directional licking in response to visual or tactile stimulus. Depending on the task rule, the mice must respond to one stimulus modality while ignoring the other. Neural activity to the same tactile stimulus is modulated by task in all the areas recorded, with significant activity changes in a subset of neurons and population activity occupying distinct activity subspaces. Recordings further reveal a contextual signal in the pre-stimulus baseline activity that differentiates task context. This signal is correlated with subsequent task modulation of neural activity. Comparison across brain areas shows that this contextual signal is stronger in frontal cortical regions than sensory regions. Analyses link this signal to behavior by showing that it tracks the behavioral performance switch during task rule transitions. Silencing activity in frontal cortical regions during the baseline period impairs behavioral performance.

Strengths:

This is a carefully done study with solid results and thorough controls. The authors identify a contextual signal in baseline neural activity that predicts rule-dependent decision-related activity. The comprehensive characterization across a sensorimotor pathway is another strength. Analyses and perturbation experiments link this contextual signal to animals' behavior. The results provide a neural substrate that will surely inspire follow-up mechanistic investigations.

Weaknesses:

None. The authors have further improved the manuscript during the revision with additional analyses.

Impact:

This study reports an important neural signature for context-dependent decision-making that has important implications for mechanisms of context-dependent neural computation in general.

---

## [Author Response]

The following is the authors’ response to the original reviews.

**Public Reviews:**

**Reviewer #1 (Public Review):**
Summary:Using a cross-modal sensory selection task in head-fixed mice, the authors attempted to characterize how different rules reconfigured representations of sensory stimuli and behavioral reports in sensory (S1, S2) and premotor cortical areas (medial motor cortex or MM, and ALM). They used silicon probe recordings during behavior, a combination of single-cell and population-level analyses of neural data, and optogenetic inhibition during the task.Strengths:A major strength of the manuscript was the clarity of the writing and motivation for experiments and analyses. The behavioral paradigm is somewhat simple but well-designed and wellcontrolled. The neural analyses were sophisticated, clearly presented, and generally supported the authors' interpretations. The statistics are clearly reported and easy to interpret. In general, my view is that the authors achieved their aims. They found that different rules affected preparatory activity in premotor areas, but not sensory areas, consistent with dynamical systems perspectives in the field that hold that initial conditions are important for determining trial-based dynamics.Weaknesses:The manuscript was generally strong. The main weakness in my view was in interpreting the optogenetic results. While the simplicity of the task was helpful for analyzing the neural data, I think it limited the informativeness of the perturbation experiments. The behavioral read-out was low dimensional -a change in hit rate or false alarm rate- but it was unclear what perceptual or cognitive process was disrupted that led to changes in these read-outs. This is a challenge for the field, and not just this paper, but was the main weakness in my view. I have some minor technical comments in the recommendations for authors that might address other minor weaknesses.

I think this is a well-performed, well-written, and interesting study that shows differences in rule representations in sensory and premotor areas and finds that rules reconfigure preparatory activity in the motor cortex to support flexible behavior.

**Reviewer #2 (Public Review):**
Summary:Chang et al. investigate neuronal activity firing patterns across various cortical regions in an interesting context-dependent tactile vs visual detection task, developed previously by the authors (Chevee et al., 2021; doi: 10.1016/j.neuron.2021.11.013). The authors report the important involvement of a medial frontal cortical region (MM, probably a similar location to wM2 as described in Esmaeili et al., 2021 & 2022; doi: 10.1016/j.neuron.2021.05.005; doi: 10.1371/journal.pbio.3001667) in mice for determining task rules.Strengths:The experiments appear to have been well carried out and the data well analysed. The manuscript clearly describes the motivation for the analyses and reaches clear and well-justified conclusions. I find the manuscript interesting and exciting!Weaknesses:I did not find any major weaknesses.
**Reviewer #3 (Public Review):**
This study examines context-dependent stimulus selection by recording neural activity from several sensory and motor cortical areas along a sensorimotor pathway, including S1, S2, MM, and ALM. Mice are trained to either withhold licking or perform directional licking in response to visual or tactile stimulus. Depending on the task rule, the mice have to respond to one stimulus modality while ignoring the other. Neural activity to the same tactile stimulus is modulated by task in all the areas recorded, with significant activity changes in a subset of neurons and population activity occupying distinct activity subspaces. Recordings further reveal a contextual signal in the pre-stimulus baseline activity that differentiates task context. This signal is correlated with subsequent task modulation of stimulus activity. Comparison across brain areas shows that this contextual signal is stronger in frontal cortical regions than in sensory regions. Analyses link this signal to behavior by showing that it tracks the behavioral performance switch during task rule transitions. Silencing activity in frontal cortical regions during the baseline period impairs behavioral performance.Overall, this is a superb study with solid results and thorough controls. The results are relevant for context-specific neural computation and provide a neural substrate that will surely inspire follow-up mechanistic investigations. We only have a couple of suggestions to help the authors further improve the paper.(1) We have a comment regarding the calculation of the choice CD in Fig S3. The text on page 7 concludes that "Choice coding dimensions change with task rule". However, the motor choice response is different across blocks, i.e. lick right vs. no lick for one task and lick left vs. no lick for the other task. Therefore, the differences in the choice CD may be simply due to the motor response being different across the tasks and not due to the task rule per se. The authors may consider adding this caveat in their interpretation. This should not affect their main conclusion.

We thank the Reviewer for the suggestion. We have discussed this caveat and performed a new analysis to calculate the choice coding dimensions using right-lick and left-lick trials (Fig. S3h) on page 8.

“Choice coding dimensions were obtained from left-lick and no-lick trials in respond-to-touch blocks and right-lick and no-lick trials in respond-to-light blocks. Because the required lick directions differed between the block types, the difference in choice CDs across task rules (Fig. S4f) could have been affected by the different motor responses. To rule out this possibility, we did a new version of this analysis using right-lick and left-lick trials to calculate the choice coding dimensions for both task rules. We found that the orientation of the choice coding dimension in a respond-to-touch block was still not aligned well with that in a respond-to-light block (Fig. S4h; magnitude of dot product between the respond-to-touch choice CD and the respond-to-light choice CD, mean ± 95% CI for true vs shuffled data: S1: 0.39 ± [0.23, 0.55] vs 0.2 ± [0.1, 0.31], 10 sessions; S2: 0.32 ± [0.18, 0.46] vs 0.2 ± [0.11, 0.3], 8 sessions; MM: 0.35 ± [0.21, 0.48] vs 0.18 ± [0.11, 0.26], 9 sessions; ALM: 0.28 ± [0.17, 0.39] vs 0.21 ± [0.12, 0.31], 13 sessions).”

We also have included the caveats for using right-lick and left-lick trials to calculate choice coding dimensions on page 13.

“However, we also calculated choice coding dimensions using only right- and left-lick trials. In S1, S2, MM and ALM, the choice CDs calculated this way were also not aligned well across task rules (Fig. S4h), consistent with the results calculated from lick and no-lick trials (Fig. S4f). Data were limited for this analysis, however, because mice rarely licked to the unrewarded water port (# of licksunrewarded port / # of lickstotal , respond-to-touch: 0.13, respond-to-light: 0.11). These trials usually came from rule transitions (Fig. 5a) and, in some cases, were potentially caused by exploratory behaviors. These factors could affect choice CDs.”

(2) We have a couple of questions about the effect size on single neurons vs. population dynamics. From Fig 1, about 20% of neurons in frontal cortical regions show task rule modulation in their stimulus activity. This seems like a small effect in terms of population dynamics. There is somewhat of a disconnect from Figs 4 and S3 (for stimulus CD), which show remarkably low subspace overlap in population activity across tasks. Can the authors help bridge this disconnect? Is this because the neurons showing a difference in Fig 1 are disproportionally stimulus selective neurons?

We thank the Reviewer for the insightful comment and agree that it is important to link the single-unit and population results. We have addressed these questions by (1) improving our analysis of task modulation of single neurons (tHit-tCR selectivity) and (2) examining the relationship between tHit-tCR selective neurons and tHit-tCR subspace overlaps.

Previously, we averaged the AUC values of time bins within the stimulus window (0-150 ms, 10 ms bins). If the 95% CI on this averaged AUC value did not include 0.5, this unit was considered to show significant selectivity. This approach was highly conservative and may underestimate the percentage of units showing significant selectivity, particularly any units showing transient selectivity. In the revised manuscript, we now define a unit as showing significant tHit-tCR selectivity when three consecutive time bins (>30 ms, 10ms bins) of AUC values were significant. Using this new criterion, the percentage of tHittCR selective neurons increased compared with the previous analysis. We have updated Figure 1h and the results on page 4:

“We found that 18-33% of neurons in these cortical areas had area under the receiver-operating curve (AUC) values significantly different from 0.5, and therefore discriminated between tHit and tCR trials (Fig. 1h; S1: 28.8%, 177 neurons; S2: 17.9%, 162 neurons; MM: 32.9%, 140 neurons; ALM: 23.4%, 256 neurons; criterion to be considered significant: Bonferroni corrected 95% CI on AUC did not include 0.5 for at least 3 consecutive 10-ms time bins).”

Next, we have checked how tHit-tCR selective neurons were distributed across sessions. We found that the percentage of tHit-tCR selective neurons in each session varied (S1: 9-46%, S2: 0-36%, MM:25-55%, ALM:0-50%). We examined the relationship between the numbers of tHit-tCR selective neurons and tHit-tCR subspace overlaps. Sessions with more neurons showing task rule modulation tended to show lower subspace overlap, but this correlation was modest and only marginally significant (r = -0.32, p = 0.08, Pearson correlation, n = 31 sessions). While we report the percentage of neurons showing significant selectivity as a simple way to summarize single-neuron effects, this does neglect the magnitude of task rule modulation of individual neurons, which may also be relevant.

In summary, the apparent disconnect between the effect sizes of task modulation of single neurons and of population dynamics could be explained by (1) the percentages of tHit-tCR selective neurons were underestimated in our old analysis, (2) tHit-tCR selective neurons were not uniformly distributed among sessions, and (3) the percentages of tHit-tCR selective neurons were weakly correlated with tHit-tCR subspace overlaps.

**Recommendations for the authors:**

**Reviewer #1 (Recommendations For The Authors):**
For the analysis of choice coding dimensions, it seems that the authors are somewhat data limited in that they cannot compare lick-right/lick-left within a block. So instead, they compare lick/no lick trials. But given that the mice are unable to initiate trials, the interpretation of the no lick trials is a bit complicated. It is not clear that the no lick trials reflect a perceptual judgment about the stimulus (i.e., a choice), or that the mice are just zoning out and not paying attention. If it's the latter case, what the authors are calling choice coding is more of an attentional or task engagement signal, which may still be interesting, but has a somewhat different interpretation than a choice coding dimension. It might be worth clarifying this point somewhere, or if I'm totally off-base, then being more clear about why lick/no lick is more consistent with choice than task engagement.

We thank the Reviewer for raising this point. We have added a new paragraph on page 13 to clarify why we used lick/no-lick trials to calculate choice coding dimensions, and we now discuss the caveat regarding task engagement.

“No-lick trials included misses, which could be caused by mice not being engaged in the task. While the majority of no-lick trials were correct rejections (respond-to-touch: 75%; respond-to-light: 76%), we treated no-licks as one of the available choices in our task and included them to calculate choice coding dimensions (Fig. S4c,d,f). To ensure stable and balanced task engagement across task rules, we removed the last 20 trials of each session and used stimulus parameters that achieved similar behavioral performance for both task rules (Fig. 1d; ~75% correct for both rules).”

In addition, to address a point made by Reviewer 3 as well as this point, we performed a new analysis to calculate choice coding dimensions using right-lick vs left-lick trials. We report this new analysis on page 8:

“Choice coding dimensions were obtained from left-lick and no-lick trials in respond-to-touch blocks and right-lick and no-lick trials in respond-to-light blocks. Because the required lick directions differed between the block types, the difference in choice CDs across task rules (Fig. S4f) could have been affected by the different motor responses. To rule out this possibility, we did a new version of this analysis using right-lick and left-lick trials to calculate the choice coding dimensions for both task rules. We found that the orientation of the choice coding dimension in a respond-to-touch block was still not aligned well with that in a respond-to-light block (Fig. S4h; magnitude of dot product between the respond-to-touch choice CD and the respond-to-light choice CD, mean ± 95% CI for true vs shuffled data: S1: 0.39 ± [0.23, 0.55] vs 0.2 ± [0.1, 0.31], 10 sessions; S2: 0.32 ± [0.18, 0.46] vs 0.2 ± [0.11, 0.3], 8 sessions; MM: 0.35 ± [0.21, 0.48] vs 0.18 ± [0.11, 0.26], 9 sessions; ALM: 0.28 ± [0.17, 0.39] vs 0.21 ± [0.12, 0.31], 13 sessions).”

We added discussion of the limitations of this new analysis on page 13:

“However, we also calculated choice coding dimensions using only right- and left-lick trials. In S1, S2, MM and ALM, the choice CDs calculated this way were also not aligned well across task rules (Fig. S4h), consistent with the results calculated from lick and no-lick trials (Fig. S4f). Data were limited for this analysis, however, because mice rarely licked to the unrewarded water port (# of licksunrewarded port / # of lickstotal , respond-to-touch: 0.13, respond-to-light: 0.11). These trials usually came from rule transitions (Fig. 5a) and, in some cases, were potentially caused by exploratory behaviors. These factors could affect choice CDs.”

The authors find that the stimulus coding direction in most areas (S1, S2, and MM) was significantly aligned between the block types. How do the authors interpret that finding? That there is no major change in stimulus coding dimension, despite the change in subspace? I think I'm missing the big picture interpretation of this result.

That there is no significant change in stimulus coding dimensions but a change in subspace suggests that the subspace change largely reflects a change in the choice coding dimensions.

As I mentioned in the public review, I thought there was a weakness with interpretation of the optogenetic experiments, which the authors generally interpret as reflecting rule sensitivity. However, given that they are inhibiting premotor areas including ALM, one might imagine that there might also be an effect on lick production or kinematics. To rule this out, the authors compare the change in lick rate relative to licks during the ITI. What is the ITI lick rate? I assume pretty low, once the animal is welltrained, in which case there may be a floor effect that could obscure meaningful effects on lick production. In addition, based on the reported CI on delta p(lick), it looks like MM and AM did suppress lick rate. I think in the future, a task with richer behavioral read-outs (or including other measurements of behavior like video), or perhaps something like a psychological process model with parameters that reflect different perceptual or cognitive processes could help resolve the effects of perturbations more precisely.

Eighteen and ten percent of trials had at least one lick in the ITI in respond-to-touch and respond-tolight blocks, respectively. These relatively low rates of ITI licking could indeed make an effect of optogenetics on lick production harder to observe. We agree that future work would benefit from more complex tasks and measurements, and have added the following to make this point (page 14):

“To more precisely dissect the effects of perturbations on different cognitive processes in rule-dependent sensory detection, more complex behavioral tasks and richer behavioral measurements are needed in the future.”

**Reviewer #2 (Recommendations For The Authors):**
I have the following minor suggestions that the authors might consider in revising this already excellent manuscript :(1) In addition to showing normalised z-score firing rates (e.g. Fig 1g), I think it is important to show the grand-average mean firing rates in Hz.

We thank the Reviewer for the suggestion and have added the grand-average mean firing rates as a new supplementary figure (Fig. S2a). To provide more details about the firing rates of individual neurons, we have also added to this new figure the distribution of peak responses during the tactile stimulus period (Fig. S2b).

(2) I think the authors could report more quantitative data in the main text. As a very basic example, I could not easily find how many neurons, sessions, and mice were used in various analyses.

We have added relevant numbers at various points throughout the Results, including within the following examples:

Page 3: “To examine how the task rules influenced the sensorimotor transformation occurring in the tactile processing stream, we performed single-unit recordings from sensory and motor cortical areas including S1, S2, MM and ALM (Fig. 1e-g, Fig. S1a-h, and Fig. S2a; S1: 6 mice, 10 sessions, 177 neurons, S2: 5 mice, 8 sessions, 162 neurons, MM: 7 mice, 9 sessions, 140 neurons, ALM: 8 mice, 13 sessions, 256 neurons).”

Page 5: “As expected, single-unit activity before stimulus onset did not discriminate between tactile and visual trials (Fig. 2d; S1: 0%, 177 neurons; S2: 0%, 162 neurons; MM: 0%, 140 neurons; ALM: 0.8%, 256 neurons). After stimulus onset, more than 35% of neurons in the sensory cortical areas and approximately 15% of neurons in the motor cortical areas showed significant stimulus discriminability (Fig. 2e; S1: 37.3%, 177 neurons; S2: 35.2%, 162 neurons; MM: 15%, 140 neurons; ALM: 14.1%, 256 neurons).”

Page 6: “Support vector machine (SVM) and Random Forest classifiers showed similar decoding abilities

(Fig. S3a,b; medians of classification accuracy [true vs shuffled]; SVM: S1 [0.6 vs 0.53], 10 sessions, S2 [0.61 vs 0.51], 8 sessions, MM [0.71 vs 0.51], 9 sessions, ALM [0.65 vs 0.52], 13 sessions; Random Forests: S1 [0.59 vs 0.52], 10 sessions, S2 [0.6 vs 0.52], 8 sessions, MM [0.65 vs 0.49], 9 sessions, ALM [0.7 vs 0.5], 13 sessions).”

Page 6: “To assess this for the four cortical areas, we quantified how the tHit and tCR trajectories diverged from each other by calculating the Euclidean distance between matching time points for all possible pairs of tHit and tCR trajectories for a given session and then averaging these for the session (Fig. 4a,b; S1: 10 sessions, S2: 8 sessions, MM: 9 sessions, ALM: 13 sessions, individual sessions in gray and averages across sessions in black; window of analysis: -100 to 150 ms relative to stimulus onset; 10 ms bins; using the top 3 PCs; Methods).”

Page 8: “In contrast, we found that S1, S2 and MM had stimulus CDs that were significantly aligned between the two block types (Fig. S4e; magnitude of dot product between the respond-to-touch stimulus CDs and the respond-to-light stimulus CDs, mean ± 95% CI for true vs shuffled data: S1: 0.5 ± [0.34, 0.66] vs 0.21 ± [0.12, 0.34], 10 sessions; S2: 0.62 ± [0.43, 0.78] vs 0.22 ± [0.13, 0.31], 8 sessions; MM: 0.48 ± [0.38, 0.59] vs 0.24 ± [0.16, 0.33], 9 sessions; ALM: 0.33 ± [0.2, 0.47] vs 0.21 ± [0.13, 0.31], 13 sessions).” Page 9: “For respond-to-touch to respond-to-light block transitions, the fractions of trials classified as respond-to-touch for MM and ALM decreased progressively over the course of the transition (Fig. 5d; rank correlation of the fractions calculated for each of the separate periods spanning the transition, Kendall’s tau, mean ± 95% CI: MM: -0.39 ± [-0.67, -0.11], 9 sessions, ALM: -0.29 ± [-0.54, -0.04], 13 sessions; criterion to be considered significant: 95% CI on Kendall’s tau did not include 0).

Page 11: “Lick probability was unaffected during S1, S2, MM and ALM experiments for both tasks, indicating that the behavioral effects were not due to an inability to lick (Fig. 6i, j; 95% CI on Δ lick probability for cross-modal selection task: S1/S2 [-0.18, 0.24], 4 mice, 10 sessions; MM [-0.31, 0.03], 4 mice, 11 sessions; ALM [-0.24, 0.16], 4 mice, 10 sessions; Δ lick probability for simple tactile detection task: S1/S2 [-0.13, 0.31], 3 mice, 3 sessions; MM [-0.06, 0.45], 3 mice, 5 sessions; ALM [-0.18, 0.34], 3 mice, 4 sessions).”

(3) Please include a clearer description of trial timing. Perhaps a schematic timeline of when stimuli are delivered and when licking would be rewarded. I may have missed it, but I did not find explicit mention of the timing of the reward window or if there was any delay period.

We have added the following (page 3):

“For each trial, the stimulus duration was 0.15 s and an answer period extended from 0.1 to 2 s from stimulus onset.”

(4) Please include a clear description of statistical tests in each figure legend as needed (for example please check Fig 4e legend).

We have added details about statistical tests in the figure legends:

Fig. 2f: “Relationship between block-type discriminability before stimulus onset and tHit-tCR discriminability after stimulus onset for units showing significant block-type discriminability prior to the stimulus. Pearson correlation: S1: r = 0.69, p = 0.056, 8 neurons; S2: r = 0.91, p = 0.093, 4 neurons; MM: r = 0.93, p < 0.001, 30 neurons; ALM: r = 0.83, p < 0.001, 26 neurons.”

Fig. 4e: “Subspace overlap for control tHit (gray) and tCR (purple) trials in the somatosensory and motor cortical areas. Each circle is a subspace overlap of a session. Paired t-test, tCR – control tHit: S1: -0.23, 8 sessions, p = 0.0016; S2: -0.23, 7 sessions, p = 0.0086; MM: -0.36, 5 sessions, p = <0.001; ALM: -0.35, 11 sessions, p < 0.001; significance: ** for p<0.01, *** for p<0.001.”

Fig. 5d,e: “Fraction of trials classified as coming from a respond-to-touch block based on the pre-stimulus population state, for trials occurring in different periods (see **c**) relative to respond-to-touch → respondto-light transitions. For MM (top row) and ALM (bottom row), progressively fewer trials were classified as coming from the respond-to-touch block as analysis windows shifted later relative to the rule transition. Kendall’s tau (rank correlation): MM: -0.39, 9 sessions; ALM: -0.29, 13 sessions. Left panels: individual sessions, right panels: mean ± 95% CI. Dash lines are chance levels (0.5). **e**, Same as **d** but for respond-to-light → respond-to-touch transitions. Kendall’s tau: MM: 0.37, 9 sessions; ALM: 0.27, 13 sessions.”

Fig. 6: “Error bars show bootstrap 95% CI. Criterion to be considered significant: 95% CI did not include 0.”

(5) P. 3 - "To examine how the task rules influenced the sensorimotor transformation occurring in the tactile processing stream, we performed single-unit recordings from sensory and motor cortical areas including S1, S2, MM, and ALM using 64-channel silicon probes (Fig. 1e-g and Fig. S1a-h)." Please specify if these areas were recorded simultaneously or not.

We have added *“We recorded from one of these cortical areas per session, using 64-channel silicon probes.”* on page 3.

(6) Figure 4b - Please describe what gray and black lines show.

The gray traces are the distance between tHit and tCR trajectories in individual sessions and the black traces are the averages across sessions in different cortical areas. We have added this information on page 6 and in the Figure 4b legend.

Page 6: “To assess this for the four cortical areas, we quantified how the tHit and tCR trajectories diverged from each other by calculating the Euclidean distance between matching time points for all possible pairs of tHit and tCR trajectories for a given session and then averaging these for the session (Fig. 4a,b; S1: 10 sessions, S2: 8 sessions, MM: 9 sessions, ALM: 13 sessions, individual sessions in gray and averages across sessions in black; window of analysis: -100 to 150 ms relative to stimulus onset; 10 ms bins; using the top 3 PCs; Methods).

Fig. 4b: “Distance between tHit and tCR trajectories in S1, S2, MM and ALM. Gray traces show the time varying tHit-tCR distance in individual sessions and black traces are session-averaged tHit-tCR distance (S1:10 sessions; S2: 8 sessions; MM: 9 sessions; ALM: 13 sessions).”

(7) In addition to the analyses shown in Figure 5a, when investigating the timing of the rule switch, I think the authors should plot the left and right lick probabilities aligned to the timing of the rule switch time on a trial-by-trial basis averaged across mice.

We thank the Reviewer for suggesting this addition. We have added a new figure panel to show the probabilities of right- and left-licks during rule transitions (Fig. 5a).

Page 8: “The probabilities of right-licks and left-licks showed that the mice switched their motor responses during block transitions depending on task rules (Fig. 5a, mean ± 95% CI across 12 mice).”

(8) P. 12 - "Moreover, in a separate study using the same task (Finkel et al., unpublished), high-speed video analysis demonstrated no significant differences in whisker motion between respond-to-touch and respond-to-light blocks in most (12 of 14) behavioral sessions.". Such behavioral data is important and ideally would be included in the current analysis. Was high-speed videography carried out during electrophysiology in the current study?

Finkel et al. has been accepted in principle for publication and will be available online shortly. Unfortunately we have not yet carried out simultaneous high-speed whisker video and electrophysiology in our cross-modal sensory selection task.

**Reviewer #3 (Recommendations For The Authors):**
(1) Minor point. For subspace overlap calculation of pre-stimulus activity in Fig 4e (light purple datapoints), please clarify whether the PCs for that condition were constructed in matched time windows. If the PCs are calculated from the stimulus period 0-150ms, the poor alignment could be due to mismatched time windows.

We thank the Reviewer for the comment and clarify our analysis here. We previously used timematched windows to calculate subspace overlaps. However, the pre-stimulus activity was much weaker than the activity during the stimulus period, so the subspaces of reference tHit were subject to noise and we were not able to obtain reliable PCs. This caused the subspace overlap values between the reference tHit and control tHit to be low and variable (mean ± SD, S1: 0.46± 0.26, n = 8 sessions, S2: 0.46± 0.18, n = 7 sessions, MM: 0.44± 0.16, n = 5 sessions, ALM: 0.38± 0.22, n = 11 sessions). Therefore, we used the tHit activity during the stimulus window to obtain PCs and projected pre-stimulus and stimulus activity in tCR trials onto these PCs. We have now added a more detailed description of this analysis in the Methods (page 32).

“To calculate the separation of subspaces prior to stimulus delivery, pre-stimulus activity in tCR trials (100 to 0 ms from stimulus onset) was projected to the PC space of the tHit reference group and the subspace overlap was calculated. In this analysis, we used tHit activity during stimulus delivery (0 to 150 ms from stimulus onset) to obtain reliable PCs.”

We acknowledge this time alignment issue and have now removed the reported subspace overlap between tHit and tCR during the pre-stimulus period from Figure 4e (light purple). However, we think the correlation between pre- and post- stimulus-onset subspace overlaps should remain similar regardless of the time windows that we used for calculating the PCs. For the PCs calculated from the pre-stimulus period (-100 to 0 ms), the correlation coefficient was 0.55 (Pearson correlation, p <0.01, n = 31 sessions). For the PCs calculated from the stimulus period (0-150 ms), the correlation coefficient was 0.68 (Figure 4f, Pearson correlation, p <0.001, n = 31 sessions). Therefore, we keep Figure 4f.

(2) Minor point. To help the readers follow the logic of the experiments, please explain why PPC and AMM were added in the later optogenetic experiment since these are not part of the electrophysiology experiment.

We have added the following rationale on page 9.

“We recorded from AMM in our cross-modal sensory selection task and observed visually-evoked activity (Fig. S1i-k), suggesting that AMM may play an important role in rule-dependent visual processing. PPC contributes to multisensory processing51–53 and sensory-motor integration50,54–58. Therefore, we wanted to test the roles of these areas in our cross-modal sensory selection task.”

(3) Minor point. We are somewhat confused about the timing of some of the example neurons shown in figure S1. For example, many neurons show visually evoked signals only after stimulus offset, unlike tactile evoked signals (e.g. Fig S1b and f). In addition, the reaction time for visual stimulus is systematically slower than tactile stimuli for many example neurons (e.g. Fig S1b) but somehow not other neurons (e.g. Fig S1g). Are these observations correct?

These observations are all correct. We have a manuscript from a separate study using this same behavioral task (Finkel et al., accepted in principle) that examines and compares (1) the onsets of tactile- and visually-evoked activity and (2) the reaction times to tactile and visual stimuli. The reaction times to tactile stimuli were slightly but significantly shorter than the reaction times to visual stimuli (tactile vs visual, 397 ± 145 vs 521 ± 163 ms, median ± interquartile range [IQR], Tukey HSD test, p = 0.001, n = 155 sessions). We examined how well activity of individual neurons in S1 could be used to discriminate the presence of the stimulus or the response of the mouse. For discriminability for the presence of the stimulus, S1 neurons could signal the presence of the tactile stimulus but not the visual stimulus. For discriminability for the response of the mouse, the onsets for significant discriminability occurred earlier for tactile compared with visual trials (two-sided Kolmogorov-Smirnov test, p = 1x10-16, n = 865 neurons with DP onset in tactile trials, n = 719 neurons with DP onset in visual trials).